# CHUNKING THE CRITIC: A TRANSFORMER-BASED SOFT ACTOR-CRITIC WITH N-STEP RETURNS

**Dong Tian**[*]  **Onur Celik**  **Gerhard Neumann**
Karlsruhe Institute of Technology (KIT)

## ABSTRACT

We introduce a sequence-conditioned critic for Soft Actor–Critic (SAC) that models trajectory context with a lightweight Transformer and trains on aggregated $N$-step targets. Unlike prior approaches that (i) score state–action pairs in isolation or (ii) rely on actor-side action chunking to handle long horizons, our method strengthens the critic itself by conditioning on short trajectory segments and integrating multi-step returns without the need of importance sampling (IS). The resulting sequence-aware value estimates capture the critical temporal structure for extended-horizon and sparse-reward problems. On multiple benchmarks, we further show that freezing critic parameters for several steps makes our update compatible with CrossQ's core idea, enabling stable training *without* a target network. Despite its simplicity, a 2-layer Transformer with $128$–$256$ hidden units and a maximum update-to-data ratio (UTD) of $1$, the approach consistently outperforms standard SAC and strong off-policy baselines, with particularly large gains on long-trajectory control. These results highlight the value of sequence modeling and $N$-step bootstrapping on the critic side for long-horizon reinforcement learning. Code is available on GitHub, and full test logs can be viewed on Weights & Biases.

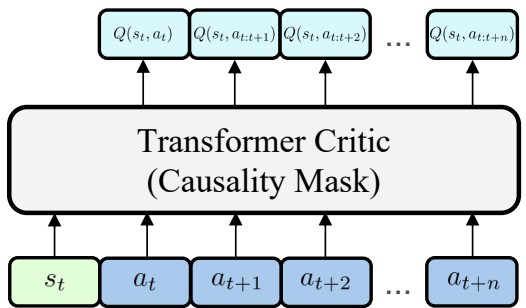

**(a)** Transformer critic processes segments of actions rather than a single action, using causal self-attention so token $i$ attends only to timesteps $\leq i$, preventing future-information leakage.

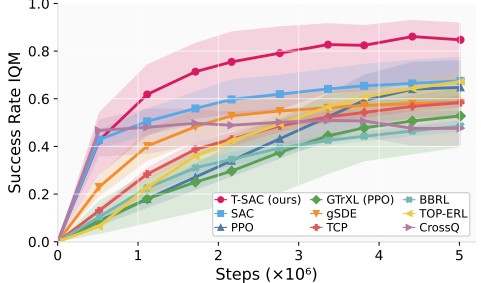

**(b)** Aggregated success on Meta–World ML1 (50 tasks) comparing T–SAC to step-based and episodic baselines. Curves show IQM success rate with 95% bootstrap confidence intervals; unless noted, results average 8 seeds and count success only at the final timestep.

**Figure 1:** T–SAC overview and aggregate Meta-World ML1 results.

## 1 INTRODUCTION

Off–policy actor–critic methods are the workhorses of continuous control. Soft Actor–Critic (SAC) (Haarnoja et al., 2018a) is notable for its sample efficiency and stability, driven by tightly controlled bootstrap targets and mechanisms that mitigate value overestimation.

---

[*]Corresponding author: dong.tian@outlook.de. This work was conducted at KIT and supported solely by the Institute for Anthropomatics and Robotics (IAR) through the Autonomous Learning Robots (ALR) Lab.

Yet accurate value estimation remains difficult in sparse-reward, long-horizon, and high-dimensional settings. Towards this end, multi-step targets ($N$-step returns) (Sutton et al., 1998) can reduce bootstrapping bias and accelerate credit assignment, but in off-policy settings they typically require importance sampling (IS) (Precup et al., 2000). IS introduces high variance and often destabilizes training in practice, which limits the effective horizon of multi-step supervision in modern actor–critic methods (Munos et al., 2016).

More recently, critics and policies have been extended to operate on *temporally extended actions*, including movement primitives (Otto et al., 2023a; Li et al., 2024b) and action chunking (Zhang et al., 2022; Li et al., 2025). By evaluating or executing short open-loop action sequences, these methods can improve exploration and speed up value propagation. However, in online reinforcement learning they introduce important trade-offs: fixing a chunk length reduces control frequency and reactivity, couples performance to a horizon hyperparameter, and complicates learning when different phases of a task benefit from different temporal resolutions. As a result, action-chunking methods have not consistently produced robust gains in online, off-policy settings, despite their success in structured or episodic domains.

Recent work has shown that the stability of off-policy actor–critic methods can be substantially improved through careful control of critic optimization dynamics. Techniques such as slowing critic updates, partial parameter freezing, and function-class regularization reduce target drift and mitigate overestimation (Vincent et al., 2025; Piché et al., 2021; Gallici et al., 2024). In this vein, CrossQ (Bhatt et al., 2019) demonstrates that appropriate normalization—via Batch Renormalization (BRN) (Ioffe, 2017) combined with bounded activations—can stabilize bootstrapped value learning to the extent that target networks can be removed altogether. These results suggest that scaling up off-policy RL does not fundamentally require architectural changes to the policy, but rather careful design of the critic's update and normalization schemes.

**This paper: Transformer-based Soft Actor-Critic (T-SAC).** Our primary contribution is T-SAC, a step-based Soft Actor-Critic in which the standard MLP critic is replaced by a sequence-conditioned Transformer critic. The critic is trained on short trajectory segments and supervised with variable-horizon N-step returns. For each segment, it predicts values for all state–action prefixes and is jointly trained across multiple temporal horizons. By modeling temporal structure inside the critic through attention over brief state–action windows, T-SAC improves long-horizon credit assignment while keeping the policy strictly one-step and the update rule free of importance sampling.

Around this core design, we introduce several supporting choices that improve stability and practicality. First, we use causal masking and a lightweight Transformer architecture, following TOP-ERL (Li et al., 2024a) in spirit but adapted to a step-based SAC setting. Second, we employ a gradient-averaged N-step loss. Instead of averaging N-step targets, which can attenuate sparse long-horizon rewards, we compute a loss for each horizon and average their gradients. Because adjacent horizons correspond to overlapping prefixes and produce correlated gradient estimates, this reduces update variance while preserving long-horizon signal. Third, we adopt a lightweight critic parameter-freezing schedule that enables stable training without target networks, in contrast to Polyak averaging as used in the original SAC (Haarnoja et al., 2018a). Empirically, T-SAC preserves SAC-style stability and remains sample efficient, solving most Meta-World tasks in approximately 5M interactions, achieving 96.8 percent success on dense Box-Pushing, and remaining stable at low update-to-data ratios across benchmarks.

## 2 RELATED WORK

### 2.1 TRANSFORMERS FOR RL

**Transformer critics for episodic vs. step-based control.** Episodic RL (ERL) replaces per-step actions with trajectory-level primitives (Otto et al., 2023a; Li et al., 2024b), easing long-horizon reasoning but complicating temporal credit assignment, especially off-policy. TOP-ERL (Li et al., 2024a) partitions each episode into fixed-length segments and trains a Transformer critic that attends across segments. With truncated $N$-step targets (Sutton et al., 1998), the critic predicts per-segment returns, enabling off-policy replay while exploiting attention for partial observability and long-range dependencies. On manipulation benchmarks, TOP-ERL improves over prior ERL-based, step-based, and on-policy value-based baselines (Li et al., 2024a).

TOP-ERL (Li et al., 2024a) is an early Transformer-critic method for *episodic* control: a ProDMP policy (Li et al., 2023) outputs full trajectories, and the critic evaluates segment-level returns along them. Replanning is discussed but not implemented in the released experiments. The method additionally relies on a Trust Region Projection Layer (TRPL) (Otto et al., 2021), typically uses ∼20M interactions, and still underperforms on some multi-phase Meta-World tasks (Yu et al., 2020) (e.g., *Assembly*, *Disassemble*) and Box–Pushing at tight tolerances (Otto et al.).

By contrast, T-SAC remains in the standard step-based, closed-loop regime: the policy outputs an action at every time step from the current state, and the Transformer critic is *prefix-conditioned* on short state–action windows sampled from replay. It is trained with non-soft $N$-step TD targets *without* importance sampling, keeping it closer to conventional off-policy actor–critic methods than to episodic ERL: temporal abstraction lives in the critic's conditioning and targets, not in an open-loop policy.

**Transformer policies for offline RL.** Decision Transformer and related offline sequence-modeling approaches (Chen et al., 2021; Janner et al., 2021) instead perform *policy-side* sequence modeling on fixed datasets, mapping past trajectories and target returns directly to actions. These methods are complementary to T-SAC: we use a Transformer only for the critic, in an *online*, off-policy setting. In principle, an offline Decision Transformer policy could be paired with a T-SAC-style critic, or our critic architecture could be adapted to evaluate trajectories generated by such sequence policies.

### 2.2 TRAINING WITHOUT TARGET NETWORK

Target networks stabilize bootstrapped critics but slow value propagation and add complexity (Kim et al., 2019; Piché et al., 2021). Recent work instead limits target drift or smooths backups. The strongest result, **CrossQ**, achieves state-of-the-art (SOTA) sample efficiency in continuous control by removing the target network and stabilizes a single bootstrapped critic with BRN (Ioffe, 2017). Related target-free strategies include value smoothing (mellowmax) (Asadi & Littman, 2017; Kim et al., 2019), constrained/proximal updates (Durugkar & Stone, 2018; Ohnishi et al., 2019), function-space regularization and partial freezing (Piché et al., 2021; Asadi et al., 2024; Vincent et al., 2025) feature decorrelation (Mavrin et al., 2019). Theory unifies these mechanisms as alternatives to target networks via partial freezing, regularization, and separation of optimization dynamics (Fellows et al., 2023).

### 2.3 ACTION CHUNKING IN REINFORCEMENT LEARNING

Action chunking replaces per-step control with short open-loop sequences of actions ("chunks"), which can capture temporal structure, accelerate value propagation via longer effective horizons, and promote temporally coherent exploration (Kalyanakrishnan et al., 2021; Zhang et al., 2022). The trade-off is reduced reactivity within a chunk (Liu et al., 2024), but for long-horizon, sparse-reward manipulation this bias often pays off (Zhang et al., 2021; Gupta et al., 2019).

*Reinforcement Learning with Action Chunking* (Q-chunking) (Li et al., 2025) applies TD-based actor–critic learning directly in the chunked action space: the policy proposes an $H$-step action sequence and the critic evaluates $Q(s_t, a_{t:t+H-1})$, enabling unbiased $H$-step backups and efficient updates (Li et al., 2025). In their implementation, the critic is a simple MLP that ingests the state concatenated with the proposed action chunk (rather than a sequence model), which keeps the method lightweight while still reaping the benefits of temporally extended actions (Li et al., 2025).

## 3 PRELIMINARIES

**Off-Policy Reinforcement Learning.** Reinforcement learning (RL) (Sutton et al., 1998) formalizes sequential decision making as a Markov decision process (MDP) $(\mathcal{S}, \mathcal{A}, P, r, \gamma)$: at time $t$ an agent observes $s_t$, selects $a_t \sim \pi(\cdot \mid s_t)$, receives $r_t = r(s_t, a_t)$, and transitions to $s_{t+1} \sim P(\cdot \mid s_t, a_t)$; the goal is to learn a policy maximizing $J(\pi) = \mathbb{E}_{\pi,P}[\sum_{t=0}^{\infty} \gamma^t r_t]$ using value functions $V^\pi(s)$ and $Q^\pi(s, a)$ that satisfy Bellman consistency, with $Q^\star$ inducing the optimal policy. Algorithms differ in how they estimate and improve these quantities—value-based learning (Watkins & Dayan, 1992; Hessel et al., 2018; Van Hasselt et al., 2016; Rummery & Niranjan, 1994), actor–critic (Mnih et al., 2016; Schulman et al., 2017; 2015a; Fujimoto et al., 2018), or direct policy optimization (Kakade,

2001; Peters & Schaal, 2008)—while managing exploration vs. exploitation (Sutton et al., 1998). Off-policy RL learns a target policy $\pi$ from data generated by a (possibly different) behavior policy $\mu$, reusing transitions $(s, a, r, s')$ via replay buffers and bootstrapped Bellman updates; distribution mismatch when evaluating $\pi$ from $\mu$-data can be corrected (e.g., with IS (Sutton et al., 1998)). This decoupling enables efficient experience reuse and underpins methods like Q-learning (Watkins & Dayan, 1992) and the SAC (Haarnoja et al., 2018a) family.

**Soft Actor-Critic.** Let $\pi_\theta(a \mid s)$ be a stochastic policy with parameters $\theta$. Let $Q_\psi(s, a)$ be the critic with parameters $\psi$, and let $Q_\phi$ be its target network (e.g., a Polyak-averaged copy of $Q_\psi$). SAC (Haarnoja et al., 2018a) maximizes a maximum-entropy objective to improve robustness and exploration:

$$J(\pi_\theta) = \mathbb{E}\Big[ \sum_{t=0}^{\infty} \gamma^t \big(r_t + \alpha\, \mathcal{H}(\pi_\theta(\cdot \mid s_t))\big) \Big].$$

where $\gamma$ is the discount factor.

The Bellman target is

$$y_t = r_t + \gamma\, \mathbb{E}_{a' \sim \pi_\theta(\cdot \mid s_{t+1})}\big[ Q_\phi(s_{t+1}, a') - \alpha \log \pi_\theta(a' \mid s_{t+1}) \big],$$

and the critic minimizes the squared error

$$J_Q(\psi) = \mathbb{E}_{(s_t, a_t, r_t, s_{t+1}) \sim \mathcal{D}}\Big[ \tfrac{1}{2}\big(Q_\psi(s_t, a_t) - y_t\big)^2 \Big].$$

The actor minimizes

$$J_\pi(\theta) = \mathbb{E}_{s \sim \mathcal{D},\, a \sim \pi_\theta(\cdot \mid s)}\big[ \alpha \log \pi_\theta(a \mid s) - Q_\psi(s, a) \big].$$

The temperature $\alpha$ is tuned to match a target entropy $\bar{\mathcal{H}}$ by minimizing

$$J(\alpha) = \mathbb{E}_{s \sim \mathcal{D},\, a \sim \pi_\theta(\cdot \mid s)}\big[ -\alpha\big(\log \pi_\theta(a \mid s) + \bar{\mathcal{H}}\big) \big] \quad \text{(Haarnoja et al., 2018b)}.$$

Recent work studies more expressive MaxEnt actors beyond Gaussians, e.g., energy-based policies trained with Stein variational updates and practical entropy estimation (S$^2$AC; Messaoud et al.), and diffusion-based policies that optimize a lower bound on the maximum-entropy objective (DIME; (Celik et al., 2025)). These approaches are largely orthogonal to our critic-side focus.

**N-step Returns and IS.** Using the same notation as above, on-policy N-step targets speed up credit assignment (Sutton et al., 1998; Schulman et al., 2015b; Mnih et al., 2016), i.e.,

$$y_{t,\text{soft}}^{(n)} = \sum_{k=0}^{n-1} \gamma^k\, \mathbb{E}_{a_{t+k} \sim \pi_\theta}\big[ r_{t+k} - \alpha \log \pi_\theta(a_{t+k} \mid s_{t+k}) \big] + \gamma^n\, \mathbb{E}_{a \sim \pi_\theta(\cdot \mid s_{t+n})}\big[ Q_\phi(s_{t+n}, a) \big].$$

With off-policy data drawn from a behavior policy $\mu \neq \pi_\theta$, per-decision importance ratios

$$\rho_{t+k} = \frac{\pi_\theta(a_{t+k} \mid s_{t+k})}{\mu(a_{t+k} \mid s_{t+k})}$$

can be used to correct the distributional mismatch (Sutton et al., 1998), i.e.,

$$\hat{G}_{t,\text{soft}}^{(n)} = \sum_{k=0}^{n-1} \left( \gamma^k \prod_{j=0}^{k-1} \rho_{t+j} \right) \big[ r_{t+k} - \alpha \log \pi_\theta(a_{t+k} \mid s_{t+k}) \big] + \left( \gamma^n \prod_{j=0}^{n-1} \rho_{t+j} \right) Q_\phi(s_{t+n}, a_{t+n}),$$

with the convention that an empty product equals 1. When $\mu = \pi_\theta$, all $\rho$'s are 1 and $\hat{G}_t^{(n)}$ reduces to the standard N-step target. Pure IS can introduce high variance, therefore the step length $n$ cannot be chosen to be very large (Precup et al., 2000; Sutton et al., 1998; Espeholt et al., 2018).

**Averaged N-step Returns for Critic Updates.** Using N-step returns is a standard way to reduce target bias for the critic (Sutton et al., 1998). For a starting index $t$ and horizon $n \in [1, \texttt{max\_length}]$, following Zhang et al. (2022) we define the N-step target as

$$G_{non-soft}^{(n)}(s_t, a_t, \ldots, a_{t+n-1}) = \sum_{j=0}^{n-1} \gamma^j r_{t+j} + \gamma^n V_\phi(s_{t+n}), \tag{1}$$

with discount $\gamma \in (0, 1]$ and a *target* network parameterized by $\phi$. Here $V_\phi(s) := \mathbb{E}_{a \sim \pi_\theta(\cdot|s)}\big[Q_\phi(s, a)\big]$ is the (non-entropy) bootstrap value under the current policy. While larger $n$ reduces bootstrapping bias, the variance of $G^{(n)}$ typically grows with $n$ (Precup et al., 2000). A practical variance reduction is to average partial returns (Konidaris et al., 2011; Daley et al., 2024):

$$\bar{G}^{(n)} = \frac{1}{n} \sum_{i=1}^{n} G^{(i)}. \tag{2}$$

This averaging lowers the variance of the reward-sum component from $\mathcal{O}(n)$ toward roughly $\mathcal{O}(n/4)$–$\mathcal{O}(n/3)$ (decreasing with $n$, depending on reward correlations), and makes the value-estimation term decay as $1/n$; under the same assumptions as (Daley et al., 2024), the full proof appears in App. C. This motivates using multiple horizons during critic training (see § 4.2). However, in our T-SAC implementation, we do not average $N$-step returns directly, as this strategy performs poorly in sparse-reward settings (see App. F).

## 4 TRANSFORMER-BASED SOFT ACTOR-CRITIC (T-SAC)

### 4.1 N-STEP RETURNS FOR CRITIC UPDATES

#### 4.1.1 GRADIENT-LEVEL AVERAGING OF N-STEP RETURNS

**Notation.** For horizon $i$, define the prefix-conditioned online critic output $Q_\psi^{(i)} := Q_\psi(s_t, a_t, \ldots, a_{t+i-1})$. Directly averaging targets can dilute sparse reward signals (App. F). Instead, we form per-horizon losses

$$L_i(\psi) = \tfrac{1}{2}\big(Q_\psi^{(i)} - G^{(i)}\big)^2, \qquad i = 1, \ldots, n, \tag{3}$$

where a shared-weights *online* critic outputs $Q_\psi^{(i)}$ for each prefix $(s_t, a_t, \ldots, a_{t+i-1})$ ($s_t$ and $a_t$ use separate embedding layers). We then *average gradients* across horizons:

$$\nabla_\psi \bar{L} = \frac{1}{n} \sum_{i=1}^{n} \nabla_\psi L_i(\psi). \tag{4}$$

Because adjacent horizons have overlapping targets and correspond to adjacent decoder positions in the same network, their per-parameter gradient contributions are positively—but not perfectly—correlated. Averaging therefore reduces update variance while preserving sparse signals (App. D, F; Fig. 2).

#### 4.1.2 STABLE CRITIC LEARNING WITHOUT IMPORTANCE SAMPLING

Standard off-policy N-step TD presumes that post-$a_t$ actions are drawn from the current policy $\pi_\theta$, which mismatches replay generated by a behavior policy $\mu$. Per-decision IS with $\rho_{t+k} = \frac{\pi_\theta(a_{t+k}|s_{t+k})}{\mu(a_{t+k}|s_{t+k})}$ corrects this but injects high variance (Precup et al., 2000; Sutton et al., 1998; Espeholt et al., 2018).

Similarly to Li et al. (2024a), we instead change the target: the critic predicts *prefix-conditioned* values for realized prefixes from replay,

$$\{ Q_\psi(s_t, a_{t:t+i-1}) \}_{i=1}^{n},$$

with $i$-step targets

$$G^{(i)}(s_t, a_{t:t+i-1}) = \sum_{j=0}^{i-1} \gamma^j r_{t+j} + \gamma^i V_\phi(s_{t+i}), \tag{5}$$

and the loss

$$\mathcal{L}_{\text{critic}} = \mathbb{E}_{(s_t, a_{t:t+n-1}) \sim \mathcal{D}}\Big[ \tfrac{1}{n} \sum_{i=1}^{n} \big(Q_\psi(s_t, a_{t:t+i-1}) - G^{(i)}(s_t, a_{t:t+i-1})\big)^2 \Big]. \tag{6}$$

As rewards follow the *recorded* prefix $a_{t:t+i-1}$, no assumption that actions came from $\pi_\theta$ is needed, and hence, no IS is required. Only the bootstrap at $t+i$ depends on $\pi_\theta$ via $V_\phi(s_{t+i})$.

Supervising short windows with multi-horizon targets and averaging their gradients yields stable updates and preserves sparse signals, enabling "multi-step supervision, one-step policy update" *without* IS (Fig. 2, 9b).

### 4.1.3 CONNECTION TO STANDARD N-STEP TD AND THEORETICAL GUARANTEES

Equations 5–6 can be viewed as a standard multi-step TD update in an MDP where each action prefix $a_{t:t+i-1}$ is treated as an extended action. For a fixed horizon $i$, we define

$$x = (s_t, a_{t:t+i-1}),$$

use equation 5 as the $N$-step target $G^{(i)}(x)$, and minimize the squared TD error

$$\left(Q_\psi(x) - G^{(i)}(x)\right)^2,$$

exactly as in classical $N$-step Q-learning.

The key difference to off-policy $N$-step TD with importance sampling (IS) is what the critic is asked to predict. IS-corrected targets are (in principle) unbiased for $Q^\pi$, but have high variance and typically require clipping when behavior and target policies differ. Our critic instead learns the value of realized prefixes under the replay distribution.

From a theoretical perspective, conditioned on a given state $s_t$ and realized prefix $a_{t:t+i-1}$, the distribution over future rewards is fully determined by the environment dynamics and does not depend on how this prefix was generated (behavior versus target policy). Empirically this yields more stable long-horizon learning. See App. E for the formal connection to existing $N$-step TD theory.

## 4.2 CRITIC NETWORK AND OBJECTIVE

Our critic is a causal Transformer that ingests $(s_t, a_t, a_{t+1}, \ldots, a_{t+n-1})$ and outputs the $n$ prefix-conditioned values $\{Q_\psi(s_t, a_t, \ldots, a_{t+i-1})\}_{i=1}^n$ (Fig. 1). For a mini-batch of $L$ trajectories, a random start index $t \in [0, N-n]$, and horizons $i \in \{1, \ldots, n\}$ with $n$ sampled uniformly from $\{\texttt{min\_length}, \ldots, \texttt{max\_length}\}$, the training objective is the mean-squared error over all horizons:

$$\mathcal{L}(\psi) = \frac{1}{L\,n} \sum_{k=1}^{L} \sum_{i=1}^{n} \left( Q_\psi(s_t^k, a_t^k, \ldots, a_{t+i-1}^k) - G^{(i)}(s_t^k, a_t^k, \ldots, a_{t+i-1}^k) \right)^2. \tag{7}$$

During backpropagation we apply the gradient-level averaging across $\{L_i\}_{i=1}^n$ described above. This construction leverages multi-horizon targets and inherits their variance-reduction benefits without target-level signal dilution.

## 4.3 POLICY NETWORK AND OBJECTIVE

Following Ba et al. (2016); Parisotto et al. (2020) and Plappert et al. (2017), we apply Layer Normalization to the policy's hidden layers (before the nonlinearity); Plappert et al. (2017) report this configuration to be useful for continuous-control actor–critic, especially when exploration noise is injected. The objectives remain

$$J_\pi(\theta) = \mathbb{E}_{s\sim\mathcal{D},\, a\sim\pi_\theta}\left[\, \alpha \log \pi_\theta(a \mid s) - Q_\psi(s, a) \,\right], \tag{8}$$

$$J(\alpha) = \mathbb{E}_{s\sim\mathcal{D},\, a\sim\pi_\theta}\left[\, -\alpha \left(\log \pi_\theta(a \mid s) + \bar{\mathcal{H}}\right) \,\right], \tag{9}$$

with target entropy $-\bar{\mathcal{H}}$ (typically $-\dim(\mathcal{A})$) and automatic temperature tuning (Haarnoja et al., 2018b). Unlike canonical SAC, our critic does not include entropy in the target; it estimates the standard (non-soft) action-value. The policy is optimized with an entropy-regularized objective, so exploration and regularization are handled entirely by the policy. This "non-soft critic + policy-side regularization" design is also used in MPO (Abdolmaleki et al., 2018), AWR/AWAC (Peng et al., 2019; Nair et al., 2020), and IQL/IDQL (Kostrikov et al., 2021; Hansen-Estruch et al., 2023). Throughout this paper, all value targets are **standard** (non-soft) action-values; the entropy term appears only in the policy objective and is not included in the critic targets.

### 4.4 CRITIC–PARAMETER FREEZING ENABLES TARGET–FREE TRAINING

CrossQ (Bhatt et al., 2019) removes target networks via batch renormalization (Ioffe, 2017) and bounded activations. In contrast, we eliminate Polyak updates with a short *critic–freezing* schedule: at the start of each critic segment we snapshot the online critic ($\phi \leftarrow \psi$), precompute and cache bootstrap targets $V_\phi(s)$ for all windows in that segment, and then freeze this snapshot while optimizing the online critic against the cached targets for the next $K$ updates (reusing each segment across $N_c$ windows; Gymnasium MuJoCo (Towers et al., 2024): $K$=20). This lightweight decoupling curbs target drift without batch renormalization or constrained activations, and on locomotion and sparse–reward tasks (e.g., Box–Pushing–Sparse (Otto et al.)) the resulting *hard–copy* schedule yields stable training that matches or exceeds Polyak updates.

Our scheme introduces a single hyperparameter, the freezing interval $K$, i.e., the number of critic updates for which we reuse a single value snapshot $V_\phi$. Because targets are computed once per segment before we enumerate windows, the minimum effective freezing interval is the segment length $L_\mathrm{seg}$ (for locomotion tasks, $L_\mathrm{seg} = 20$). Sweeping $K \in \{20, 100, 1000, 10000\}$ on Gymnasium MuJoCo Walker2d (Fig. 4g), we observe largely stable performance with only mild degradation for the largest $K$, suggesting that segment-level target caching already provides useful stabilization and that $K$ is not a brittle hyperparameter in our setting.

## 5 EXPERIMENTS

**Positioning T–SAC.** Prior value–based RL largely splits into (i) *step–based* methods (e.g., SAC (Haarnoja et al., 2018a), CrossQ (Bhatt et al., 2019)) that dominate locomotion tasks (e.g., Ant (Towers et al., 2024)), and (ii) *episodic/trajectory–level* methods (e.g., BBRL (Otto et al., 2023a), TOP-ERL (Li et al., 2024a)) that excel on long–horizon problems (e.g., Box–Pushing (Otto et al.), Meta–World (Yu et al., 2020)). T-SAC partially narrows the gap between these regimes: it retains one–step policy updates while using a sequence–conditioned Transformer critic, and empirically matches standard SAC on locomotion benchmarks while outperforming existing Transformer–based approaches (e.g., GTrXL–style policies and TOP-ERL) on our long–horizon tasks.

**Environments and Seeds.** We evaluate T–SAC on 57 tasks spanning Meta–World ML1 (50) (Yu et al., 2020), Gymnasium MuJoCo locomotion (5) (Towers et al., 2024), and Box–Pushing (dense/sparse; 2) (Otto et al.). Meta–World probes task generalization; Gymnasium MuJoCo covers standard locomotion; and Box–Pushing stresses precise, contact–rich manipulation. Unless noted otherwise, we report means over 8 seeds (ablations use 4) with 95% bootstrap confidence intervals (Agarwal et al., 2021). Training time per 1M environment steps, compared to off–policy baselines, is shown in App. I. Baseline implementations and hyperparameters are detailed in App. K and App. L, with environment details in App. J.

### 5.1 META–WORLD RESULTS

We run Meta–World ML1 with UTD= 1, policy delay= 5, batch size 512; training time is $\sim$3 h per 1M env steps. Across 50 tasks, T–SAC solves most within $\sim$5M steps and yields stronger aggregated IQM than strong baselines (per–task curves in App. A). On the hardest multi–phase tasks (Assembly, Disassemble, Hammer, Stick–Pull) T–SAC is particularly strong (Fig. 2). In contrast, TOP–ERL (Li et al., 2024a) typically requires 20M steps to reach similar aggregates. All comparisons use 5M env steps for T–SAC, while many baselines use larger budgets (Fig. 7, 8). Success is evaluated only at the final step (App. J), and our aggregates compute IQM *per task* and then average across tasks (unlike pooled–task IQM in TOP–ERL).

### 5.2 BOX PUSHING (DENSE AND SPARSE)

We evaluate dense and sparse variants of FANCYGYM (Otto et al.) Box–Pushing with tight success tolerances (position $\pm 5$ cm, orientation $\pm 0.5$ rad). Under dense shaping, T–SAC attains **96.8%** success (Fig. 2), exceeding prior baselines ($\leq 85\%$) under the same protocol. Under sparse rewards—where these terms apply only at the terminal step—T–SAC with the hard–copy critic reaches **60%** success, compared to TOP–ERL's **70%**. Thus T–SAC is state-of-the-art on Meta-World ML1 and dense Box-Pushing, and competitive under sparse rewards.

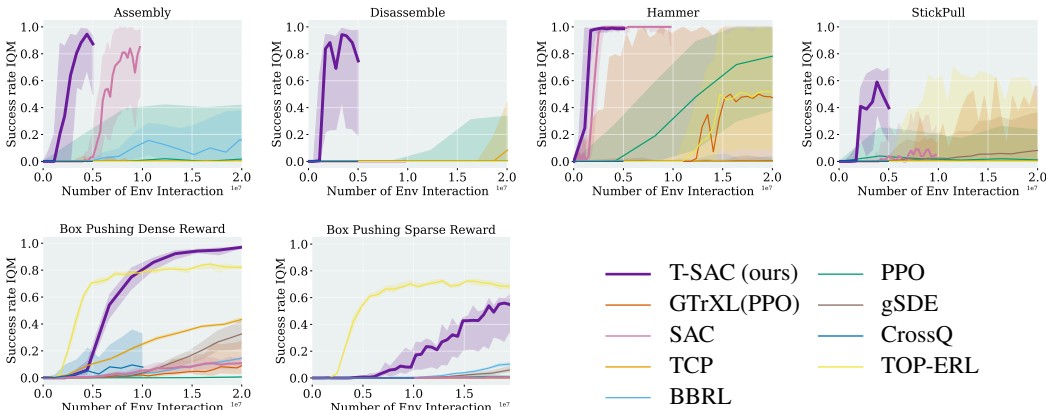

**Figure 2:** Success-rate IQM vs. environment interactions on challenging Meta-World ML1 tasks and FANCY-GYM Box-Pushing. Panels show Assembly, Disassemble, Hammer, and Stick-Pull, plus Box-Pushing under dense and sparse rewards. Success is counted only at the final timestep.

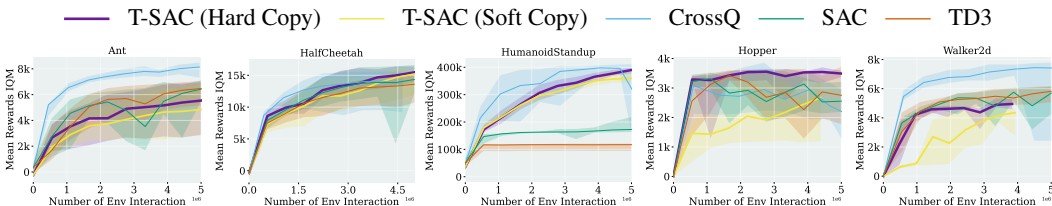

**Figure 3:** Episode return (IQM) vs. environment interactions for Ant, HalfCheetah, HumanoidStandup, Hopper, and Walker2d. Evaluation follows Gymnasium v4 native shaping/termination (no reward normalization); we report undiscounted return and use deterministic-policy evaluation.

## 5.3 GYMNASIUM MUJOCO

A lightweight *critic–parameter freezing* schedule (§ 4.4, App. B) enables target–free training: we remove the target network while retaining SAC–style stability at low update rates (UTD $\approx 0.75$) and consistently match or surpass Polyak updates. Across the five Gymnasium MuJoCo tasks, T–SAC is competitive with or better than SAC on *Ant*, *Hopper*, and *Walker2d*, with the largest gains on *HumanoidStandup* and *HalfCheetah* (Fig. 3), and we do not observe slower early convergence despite conditioning the critic on multi–step sequences from an early exploratory policy. Because episodes have variable length, we apply a simple action mask when constructing $N$-step targets from fixed–length windows to avoid bootstrapping across episode boundaries (App. H); this mask is an implementation detail rather than a core component of T–SAC and does not degrade performance.

## 5.4 ABLATION STUDY

We conduct targeted ablations on FANCYGYM Box–Pushing (dense) and MUJOCO WALKER2D. These ablations are structured to disentangle the effect of the sequence-conditioned Transformer critic—our main algorithmic contribution—from supporting design choices. Within each ablation group, all settings are identical except for the component under test; across groups, minor differences (e.g., training budget or `step_length`) arise from compute limits and are stated explicitly.

**Transformer Components.** We ablate three parts of the Transformer critic—ResNet blocks, the causal mask, and self-attention—holding all other settings fixed (Fig. 4a). Removing only self-attention invalidates the segment-conditioned objective and typically diverges. Removing self-attention *together with* the ResNet and causal mask reduces the critic to a plain MLP; we compare this baseline in § 5.4 and Fig. 4c.

**Step Length and Min Length (Reuse factor).** We sweep `step_length` and `min_length`, which bound the multi-horizon ($N$-step) supervision window. At each update we sample $n \in \{$`min_length`$, \ldots, $`step_length`$\}$; by default $n \sim \mathrm{Unif}\{1, \ldots, 16\}$. Using a *fixed* horizon

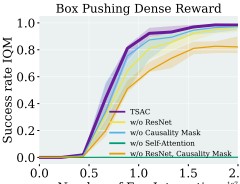

**(a)** Transformer components: effect of removing ResNet blocks, the causal mask, and/or self-attention.

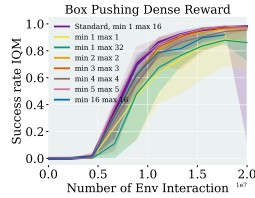

**(b)** Multi-horizon supervision window: varying (*min_length*, *max_length*) (e.g., *min* 1, *max* 16; *min* 2, *max* 2).

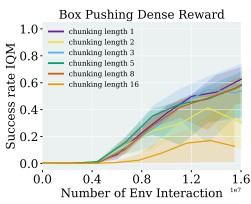

**(c)** Policy-side action chunking (MLP critic): chunk length $\in \{1, 2, 3, 5, 8, 16\}$.

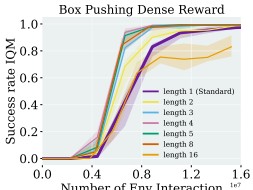

**(d)** Policy-side action chunking (Transformer critic): chunk length $\in \{1, 2, 3, 4, 5, 8, 16\}$.

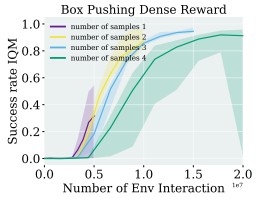

**(e)** Per-step target sampling: supervision windows per environment step $\in \{1, 2, 3, 4\}$; training budget held constant.

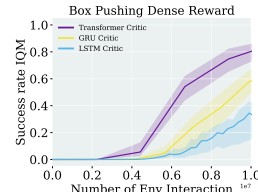

**(f)** Critic backbone: Transformer vs. GRU and LSTM.

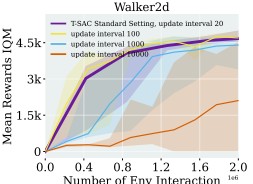

**(g)** Effect of freeze interval $K$ on Walker2d: T-SAC is stable down to $K = 20$, while very large $K$ slows learning.

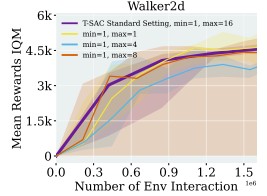

**(h)** Effect of reuse factor (`step_length`) on locomotion: moderate values are robust; larger `step_length` yields only mild gains.

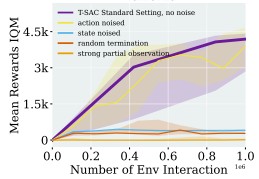

**(i)** Robustness on Walker2d under noise $\mathcal{N}(0, 0.05)$, false terminations ($p = 0.1$), and partial observability (last half of observation removed).

**Figure 4: Ablations: transformer-critic design and training settings.** Within each panel, methods share the same interaction budget and differ only in the ablated component.

$L$ smooths optimization but slightly reduces final performance (Fig. 4b), partly because the last $L-1$ states of each segment never serve as starting indices—an effect amplified for large $L$ (e.g., 16). Despite standard guidance to keep $n \leq 5$ (Precup et al., 2000; Sutton et al., 1998; Espeholt et al., 2018), our Transformer critic with gradient-level averaging is stable up to $n=16$ and benefits from longer windows (Fig. 4b). An analogous sweep under the hard-copy scheme on Gymnasium Walker2d shows consistent trends (Fig. 4h).

**Comparison to Multi-step MLP (Reinforcement Learning with Action Chunking).** With 16M environment steps (vs. our default 20M), a multi-step MLP critic consistently underperforms the Transformer critic, and policy-side chunking with an MLP yields only modest gains (Figs. 5, 4c). These policy-side baselines follow Q-Chunking (QC) (Li et al., 2025): we keep the action-chunking policy but drop the offline behavior-cloning constraint to match our online off-policy setting. In contrast, chunking the *Transformer* critic is consistently beneficial: the best configuration (chunk length 4) converges by $\approx$10M steps, reaches 99.5% final success, and shows no late-stage divergence across seeds (Fig. 4d). Since the policy is identical across QC-style settings, comparing Fig. 4c to Fig. 4d isolates the advantage of a sequence-conditioned critic under the same chunked policy. We include these results primarily to show compatibility between a Transformer critic and policy-side chunking; policy chunking itself is not our focus.

This improvement is not explained by richer input features alone. Even with single-step temporal resolution (`min_length=max_length=1`), the Transformer critic in Fig. 4b still outperforms the `chunking_length=1` baseline in Fig. 4c, suggesting the gain comes from integrating $N$-step

returns and averaging gradients over short trajectories. The MLP critic also suffers causal leakage: because it consumes fixed-length segments, $Q(s_t, a_t)$ is effectively conditioned on $(s_t, a_{t:t+n})$, i.e., it can "peek" at future actions $a_{t+1:t+n}$ (Li et al., 2025). Our Transformer critic avoids this via a causal mask (each token attends only to $t' \leq t$) and outperforms an unmasked ablation (Fig. 4a). Finally, the MLP critic requires fixed-length inputs and couples the policy chunk length to the critic, whereas the Transformer critic removes both constraints.

**Number of Samples Generated per Step.** We vary the number of supervision windows sampled per environment step ("per-step target samples"). For this study we use step_length $= [1, 8]$ (standard: $[1, 16]$). Unlike conventional SAC (one target per step), T-SAC benefits from generating multiple windows (Fig. 4e); our default is 4. In practice, use four vectorized envs or collect a single trajectory of length $4\times$max_length and slice it (App. H). Intuitively, multiple windows raise the share of fresh samples in each batch: with one window, once selected there are none left; with four, three remain.

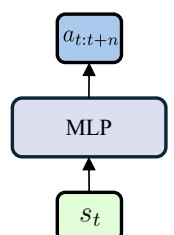

**Figure 5:** Policy chunking (adapted from Li et al. (2025)).

**GRU/LSTM as the Critic.** We replace the Transformer critic with GRU and LSTM variants under identical training (10M env steps; standard: 20M). Although recurrent critics can model action sequences, our gradient–level averaging analysis (§ 4.1.1; App. D) does not directly apply, and parallelism is reduced (Fig. 12). Empirically, both GRU and LSTM underperform the Transformer critic on Box–Pushing in our setting (Fig. 4f).

**Robustness under noise and partial observability.** We evaluate robustness to injected noise on actions and states, stochastic early termination, and partial observability via short observation windows. T-SAC degrades gracefully under action and state noise, retaining a clear performance margin over SAC and CrossQ. Stochastic termination and partial observability lead to larger drops and higher variance, but T-SAC remains at least as stable as these baselines, suggesting that sequence–conditioned critics help mitigate such effects.

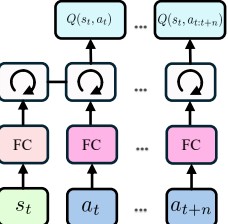

**Figure 6:** Structure of the recursive network used in our experiments.

## 6 Conclusion and Future Work

On Meta-World ML1 multiphase and FancyGym box-pushing tasks, T-SAC with a Transformer critic attains state-of-the-art success rates under fixed training budgets (5M and 20M environment steps, respectively) and a common evaluation protocol (success over 8 seeds; see §5). The sequence-conditioned critic provides smoother value estimates and more coherent long-horizon credit assignment than both largely open-loop multiphase pipelines and standard step-based value methods, yielding higher-quality continuous control.

Our study is restricted to online continuous control with low-dimensional observations. Extending T-SAC to discrete-action domains and to pixel-based or strongly partially observable settings (e.g., with visual or belief-state encoders) is nontrivial—preliminary experiments revealed instabilities and high sensitivity to architectural and optimization choices. Applying T-SAC to real-robot tasks and developing theory for when critic-side chunking provably helps, including representation analyses and shared Transformer backbones for actor–critic, are important directions for future work.

REPRODUCIBILITY STATEMENT

We made substantial efforts to ensure reproducibility. The paper and appendix detail the software/hardware environment, evaluation protocols, and all hyperparameters used. We additionally release a public GitHub repository containing the full implementation of our proposed algorithms, along with training and evaluation scripts and configuration files: here. We also provide a Weights & Biases dashboard with the complete set of test results, logs, and model artifacts: T-SAC test results. Finally, the appendix includes a step-by-step description of the experimental setup, including exact configuration details, to facilitate independent reimplementation and verification.

ETHICS STATEMENT

No human participants were involved; all data were generated in simulation. Consequently, the work does not process personal or sensitive information. We took care to avoid introducing bias in our simulated setups, but acknowledge that broader fairness and safety considerations arise in real-world deployments of such methods. We are not aware of any security, dual-use, or legal concerns stemming from this work. The authors report no conflicts of interest or external sponsorship influencing the results.

USE OF LARGE LANGUAGE MODELS (LLMS)

We used large language models (LLMs) as a general-purpose assistant during writing and development. Its roles included:

- grammar and spell-checking, language polishing, and minor stylistic edits;
- drafting and rewriting multi-paragraph text (e.g., introductions, preliminaries, and parts of experimental write-ups) based on author-provided outlines and results;
- high-level suggestions for debugging strategies and hyperparameter choices;
- assistance with literature search (proposing search queries and surfacing candidate papers).

All BibTeX entries were copied from Google Scholar; the LLMs did not generate or edit bibliographic entries. The LLMs did *not* originate the paper's main idea, problem formulation, algorithmic design, or experimental plan, and it was not used to generate or alter data, results, or figures. All citations were selected and verified by the authors against the original sources. All LLMs outputs were reviewed and, when necessary, edited or discarded. No confidential or proprietary data were shared with the LLMs.

CONTRIBUTIONS

Dong Tian led the research and implementation, including the design of the Transformer Critic, theoretical development, manuscript writing, software development, and ongoing code maintenance.

Onur Celik provided code for the experimental pipeline (e.g., CrossQ, SAC, TD3), integrating Slurm-based execution via Hydra, and sharing practical insights on state-of-the-art methods (e.g., SimbaV2, BroNet, DIME).

Gerhard Neumann initiated and supervised the project, provided mentorship and guidance in Reinforcement Learning to the lead author, proposed adapting Transformer Critics from movement-primitive-based methods to SAC, secured computational resources, and provided comprehensive feedback on the manuscript and experimental evaluations (including MetaWorld and MuJoCo).

ACKNOWLEDGMENTS

We thank Dr. Ge Li and Hongyi Zhou for insightful discussions and for sharing testing data from other projects. We thank Xin Ye for guidance on scientific writing. We are grateful to Aleksandar Taranovic for organizational support, including help in addressing computational constraints during the rebuttal phase. D.T. thanks Heiko Junker, Dr. Muhammed Türker, and Kevin Sklubal for professional guidance during his internship.

Furthermore, the first author would like to express his sincere gratitude to Gerhard Neumann and Onur Celik for their professional mentorship and for setting a standard of excellence in scientific

rigor. We also thank the anonymous reviewers on OpenReview for their constructive feedback and suggestions, which significantly improved this work.

The authors gratefully acknowledge the computing time made available to them on the high-performance computer HoreKa at the NHR Center NHR@KIT. This center is jointly supported by the Federal Ministry of Education and Research and the Ministry of Science, Research and the Arts of Baden-Württemberg, as part of the NHR joint funding program. O.C. and G.N. are supported by the European Research Council (ERC) under the European Union's Horizon Europe programme through the project SMARTI³ (Grant Agreement No. 101171393).

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

# A    APPENDIX: INDIVIDUAL META-WORLD TESTS RESULTS

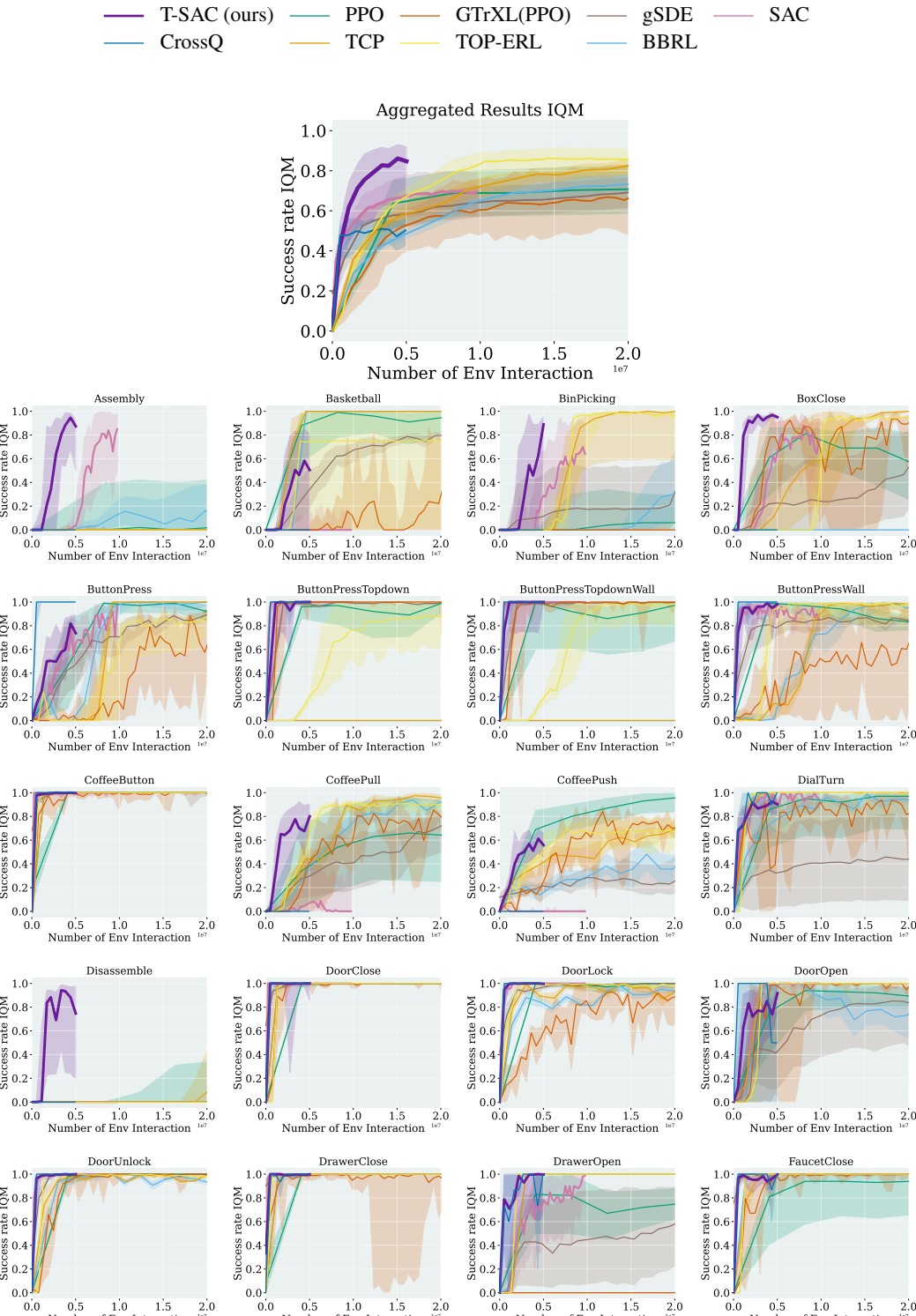

**Figure 7:** Success Rate IQM of each individual Meta-World tasks. (Part 1)

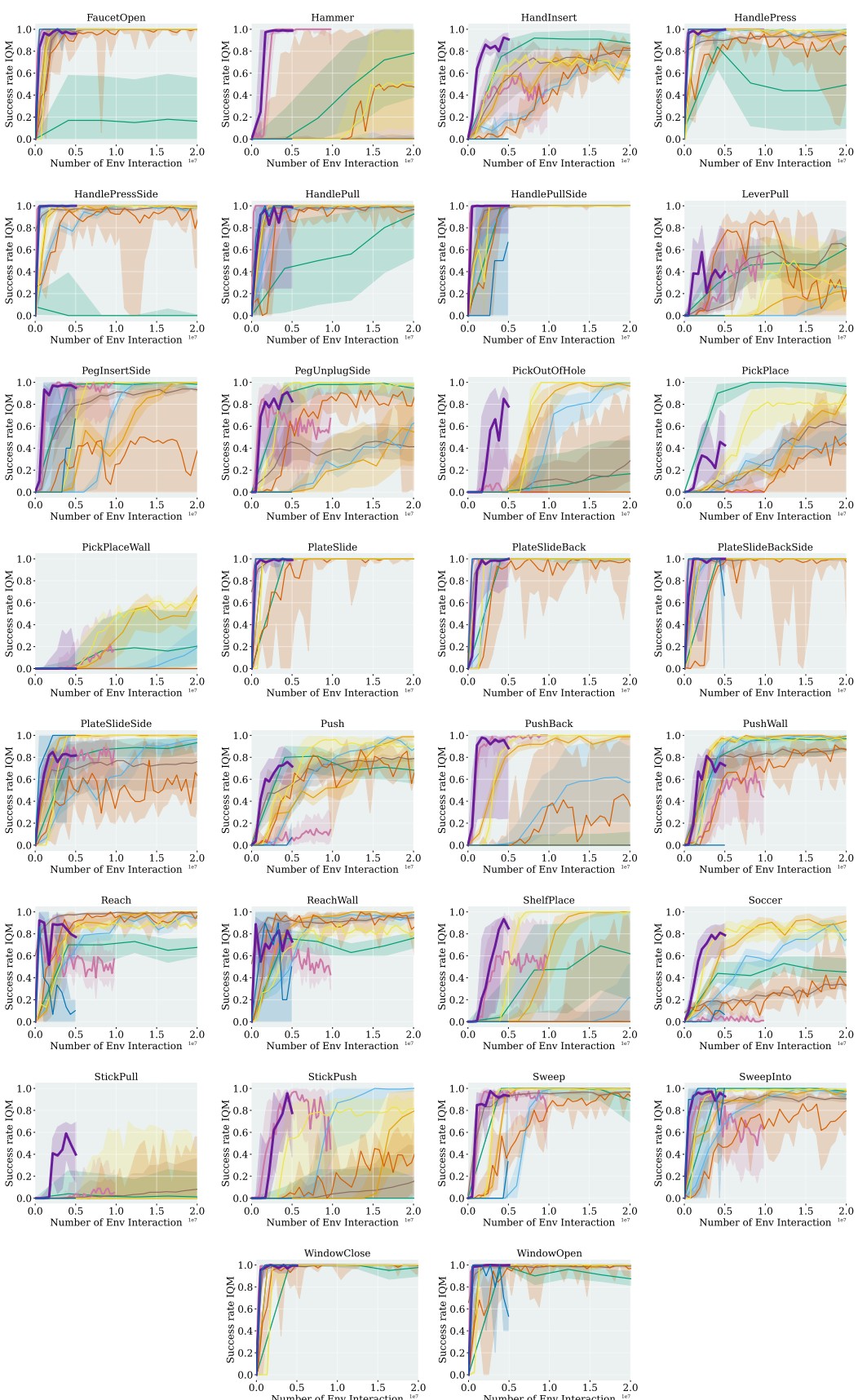

**Figure 8:** Success Rate IQM of each individual Meta-World tasks. (Part 2)

# B   APPENDIX: DETAILED ALGORITHM FLOWCHART

---

**Algorithm 1:** T-SAC

---

**Initialize:** Critic params $\phi$; target critic $\phi_{\text{target}} \leftarrow \phi$; policy params $\theta$; temperature $\alpha$; replay buffer $\mathcal{B}$; environment reset $\rightarrow s_0$.

**Input:** Segment length cap $M$; window length bounds $(\ell_{\min}, \ell_{\max})$; updates per iteration $U$; critic steps $N_c$; policy steps $N_p$; soft update $\tau$; warmups: policy / temperature.

**repeat**

    **Collect one segment (length $\leq M$)**;

    store $s_0$; $t \leftarrow 0$;

    **while** $t < M$ **do**

        sample $a_t \sim \pi_\theta(\cdot \mid s_t)$; step env $\rightarrow (r_t, s_{t+1}, d_t)$;

        append $(s_t, a_t, r_t, d_t, s_{t+1})$ to a temporary buffer;

        **if** $d_t$ **then** break;

        $t \leftarrow t+1$; $s_t \leftarrow s_{t+1}$;

    **end**

    push the whole segment $(s_{0:M}, a_{0:M-1}, r_{0:M-1}, d_{0:M-1})$ to $\mathcal{B}$;

    **if** $d_t$ **then** reset env $\rightarrow s_0$;

    **else** $s_0 \leftarrow s_M$;

    **Parameter updates**;

    **for** $u \leftarrow 1$ **to** $U$ **do**

        sample a batch of segments $\{(s_{0:M}, a_{0:M-1}, r_{0:M-1}, d_{0:M-1})\}_{b=1}^{B}$ from $\mathcal{B}$;

        precompute segment-wise bootstrapped targets using $Q_{\phi_{\text{target}}}$;

        ▷ `reuse across` $N_c$ `windows`

        **for** $k \leftarrow 1$ **to** $N_c$ **do**

            for each sequence in the batch, draw start $i$ uniformly over valid indices and draw $\ell \sim \mathcal{U}\{\ell_{\min}, \ldots, \ell_{\max}\}$ s.t. $i+\ell \leq$ segment end;

            form windows $(s_{i:i+\ell+1}, a_{i:i+\ell}, r_{i:i+\ell}, d_{i:i+\ell})$ and the corresponding $N$-step returns;

            update critic parameters $\phi$ with the Transformer critic on these windows;

            ▷ `critic update`

            $\phi_{\text{target}} \leftarrow \tau\,\phi + (1-\tau)\,\phi_{\text{target}}$;

            ▷ `soft (or hard) target update`

        **end**

        **if** *step > policy warmup* **then** sample a (fresh) batch of states from $\mathcal{B}$;

        **for** $k \leftarrow 1$ **to** $N_p$ **do**

            update policy $\theta$ by maximizing the SAC objective using $Q_\phi$;

            **if** *step > temperature warmup* **then** update temperature $\alpha$;

        **end**

        ;

    **end**

**until** *convergence*;

---

## C APPENDIX: PROOF OF THE VARIANCE REDUCTION PROPERTY OF AVERAGING OF N-STEP RETURNS

**A convenient variance identity.** Under the equicorrelation model,

$$\mathbb{E}[X_k] = m, \quad \mathrm{Var}(X_k) = v, \quad \mathrm{Cov}(X_k, X_\ell) = \rho v \ (k \neq \ell), \qquad \rho \geq 0,$$

any weighted sum $S = \sum_{k=0}^{N-1} a_k X_k$ satisfies

$$\mathrm{Var}(S) = v \left( \sum_{k=0}^{N-1} a_k^2 \ + \ \rho \left[ \left( \sum_{k=0}^{N-1} a_k \right)^2 - \sum_{k=0}^{N-1} a_k^2 \right] \right). \tag{10}$$

This follows by expanding $\mathrm{Var}$ and collecting diagonal/off-diagonal terms.

**Reward part with discount $\gamma < 1$.** Define the single $N$-step discounted reward sum and its triangular average by

$$R_N(\gamma) \triangleq \sum_{k=0}^{N-1} \gamma^k r_k, \qquad \bar{R}_N(\gamma) \triangleq \frac{1}{N} \sum_{i=1}^{N} \sum_{k=0}^{i-1} \gamma^k r_k \ = \ \sum_{k=0}^{N-1} w_k r_k, \quad w_k \triangleq \frac{N-k}{N} \gamma^k.$$

Let

$$S_0 \equiv S_0(N, \gamma) \triangleq \sum_{k=0}^{N-1} \gamma^{2k} = \frac{1 - \gamma^{2N}}{1 - \gamma^2}, \qquad T_0 \equiv T_0(N, \gamma) \triangleq \sum_{k=0}^{N-1} \gamma^k = \frac{1 - \gamma^N}{1 - \gamma}.$$

Also define the (discounted, triangular) weight aggregates

$$A_\gamma \triangleq \sum_{k=0}^{N-1} w_k^2 \ = \ \frac{1}{N^2} \sum_{k=0}^{N-1} (N-k)^2 \gamma^{2k}, \qquad B_\gamma \triangleq \left( \sum_{k=0}^{N-1} w_k \right)^2 - A_\gamma. \tag{11}$$

**Lemma 1** (Variance formulas for the discounted reward part). *Under the reward assumptions stated in the setup,*

$$\mathrm{Var}\big[R_N(\gamma)\big] = \sigma^2 \big[ S_0 + \rho\,(T_0^2 - S_0) \big], \qquad \mathrm{Var}\big[\bar{R}_N(\gamma)\big] = \sigma^2 \big[ A_\gamma + \rho\,B_\gamma \big].$$

*Proof.* Apply equation 10 with weights $a_k = \gamma^k$ for $R_N(\gamma)$ and $a_k = w_k$ for $\bar{R}_N(\gamma)$, and use the definitions of $S_0, T_0, A_\gamma, B_\gamma$. $\square$

**Proposition 1** (Reward-side variance ratio and bounds). *Define*

$$R_\gamma(N) \triangleq \frac{\mathrm{Var}[R_N(\gamma)]}{\mathrm{Var}[\bar{R}_N(\gamma)]} = \frac{S_0 + \rho\,(T_0^2 - S_0)}{A_\gamma + \rho B_\gamma}.$$

*Then for all $N \geq 1$, $\rho \geq 0$, and $\gamma \in (0, 1]$,*

$$1 \ \leq \ R_\gamma(N) \ < \ 4.$$

*Moreover, for $\gamma = 1$, $R_\gamma(N) \nearrow 4$ as $N \to \infty$; for any fixed $\gamma \in (0, 1)$, $R_\gamma(N) \to 1$. When $\rho > 0$, $R_\gamma(N)$ is strictly increasing in $N$ for $\gamma = 1$; for $\gamma < 1$ it need not be monotone.*

*Proof (sketch).* The $\gamma = 1$ proof carries through verbatim after replacing the unweighted triangular weights $j$ by the discounted weights $w_k = ((N-k)/N)\gamma^k$; the same algebraic positivity arguments yield the bounds. For the limits, when $\gamma < 1$ we have $w_k \to \gamma^k$ pointwise as $N \to \infty$ and dominated convergence gives $A_\gamma \to S_0$ and $\sum_k w_k \to T_0$, hence $R_\gamma(N) \to 1$. For $\gamma = 1$, the standard triangular-sum identities imply $R_1(N) \nearrow 4$. $\square$

**Bootstrap value part.** Let $Z_i \triangleq V_{\phi_{\mathrm{tar}}}(s_{t+i})$ denote the (target) values used for bootstrapping, and assume

$$\mathbb{E}[Z_i] = \nu, \quad \mathrm{Var}(Z_i) = \tau^2, \quad \mathrm{Cov}(Z_i, Z_j) = \kappa \tau^2 \ (i \neq j), \qquad \kappa \geq 0,$$

and that $\{r_k\}$ and $\{Z_i\}$ are independent unless stated otherwise. The single $N$-step bootstrap term and its triangular average are

$$B_N(\gamma) \triangleq \gamma^N Z_N, \qquad \bar{B}_N(\gamma) \triangleq \frac{1}{N} \sum_{i=1}^{N} \gamma^i Z_i.$$

Let

$$S_1 \equiv S_1(N, \gamma) \triangleq \sum_{i=1}^{N} \gamma^{2i} = \frac{\gamma^2(1 - \gamma^{2N})}{1 - \gamma^2}, \qquad C \equiv C(N, \gamma) \triangleq \sum_{i=1}^{N} \gamma^i = \frac{\gamma(1 - \gamma^N)}{1 - \gamma}.$$

**Lemma 2** (Variance of the averaged bootstrap part). *With $\mathrm{Cov}(Z_i, Z_j) = \kappa \tau^2$ for $i \neq j$,*

$$\mathrm{Var}[\bar{B}_N(\gamma)] = \frac{\tau^2}{N^2} \Big[ S_1 + \kappa \, (C^2 - S_1) \Big].$$

*Proof.* Apply equation 10 with $a_i = \gamma^i / N$. $\qquad\square$

**Proposition 2** (Bootstrap-side variance ratio, bounds, and condition). *Define*

$$R_B(N, \gamma, \kappa) \triangleq \frac{\mathrm{Var}[B_N(\gamma)]}{\mathrm{Var}[\bar{B}_N(\gamma)]} = \frac{N^2 \gamma^{2N}}{S_1 + \kappa(C^2 - S_1)}.$$

*Then for any $\kappa \in [0, 1]$,*

$$\frac{N^2 \gamma^{2N}}{C^2} \leq R_B(N, \gamma, \kappa) \leq \frac{N^2 \gamma^{2N}}{S_1}, \qquad \frac{\partial R_B}{\partial \kappa} < 0.$$

*In particular, averaging reduces bootstrap variance ($R_B \geq 1$) whenever*

$$\kappa \leq \kappa_\star(N, \gamma) \triangleq \frac{N^2 \gamma^{2N} - S_1}{C^2 - S_1}.$$

*For the uncorrelated case ($\kappa = 0$), $R_B(N, \gamma, 0) = N^2 \gamma^{2N} / S_1$.*

*Proof.* Monotonicity in $\kappa$ is immediate from the denominator. The bounds follow from $S_1 \leq S_1 + \kappa(C^2 - S_1) \leq C^2$. Solve $R_B \geq 1$ for $\kappa$ to get $\kappa_\star$. $\qquad\square$

**Putting the parts together.** With $G_N(\gamma) = R_N(\gamma) + B_N(\gamma)$ and $\bar{G}_N(\gamma) = \bar{R}_N(\gamma) + \bar{B}_N(\gamma)$, and assuming independence between rewards and bootstrap values,

$$\frac{\mathrm{Var}[G_N(\gamma)]}{\mathrm{Var}[\bar{G}_N(\gamma)]} = \frac{\mathrm{Var}[R_N(\gamma)] + \mathrm{Var}[B_N(\gamma)]}{\mathrm{Var}[\bar{R}_N(\gamma)] + \mathrm{Var}[\bar{B}_N(\gamma)]}. \tag{12}$$

Since all terms are nonnegative,

$$\min\big\{ R_\gamma(N), \, R_B(N, \gamma, \kappa) \big\} \leq \frac{\mathrm{Var}[G_N(\gamma)]}{\mathrm{Var}[\bar{G}_N(\gamma)]} \leq \max\big\{ R_\gamma(N), \, R_B(N, \gamma, \kappa) \big\}.$$

Consequently:

- Because $R_\gamma(N) \geq 1$ (Prop. 1), if $R_B(N, \gamma, \kappa) \geq 1$ (e.g., $\kappa \leq \kappa_\star$), then averaging N-step targets strictly reduces total variance.

- Even if $R_B(N, \gamma, \kappa) < 1$, the overall ratio in equation 12 remains $\geq 1$ whenever the reward-side gain dominates:

$$R_\gamma(N) \geq \frac{\mathrm{Var}[\bar{B}_N(\gamma)]}{\mathrm{Var}[B_N(\gamma)]} = \frac{S_1 + \kappa(C^2 - S_1)}{N^2 \gamma^{2N}}.$$

**Dependence between rewards and bootstrap values.** If $\mathrm{Cov}\big(R_N(\gamma), B_N(\gamma)\big)$ and $\mathrm{Cov}\big(\bar{R}_N(\gamma), \bar{B}_N(\gamma)\big)$ are nonzero, the numerator/denominator of equation 12 each acquire an additional covariance term. The sandwich bound above still applies after inserting these, and a crude control is $|\mathrm{Cov}(X, Y)| \leq \sqrt{\mathrm{Var}(X)\mathrm{Var}(Y)}$ (Cauchy–Schwarz), which cannot overturn the above conclusions unless the cross-covariances are pathologically large.

**Useful closed forms.** Besides $S_0, S_1, T_0, C$ above, one has

$$\sum_{k=0}^{N-1}(N-k)\gamma^k = \frac{1-(N+1)\gamma^N+N\gamma^{N+1}}{(1-\gamma)^2}.$$

A closed form for $\sum_{k=0}^{N-1}(N-k)^2\gamma^{2k}$ (hence $A_\gamma$ via equation 11) follows from the standard identities for $\sum kx^k$ and $\sum k^2 x^k$ after the change $k \mapsto N-1-k$; we omit it as not needed for the bounds above.

**Remarks.** (i) Setting $\gamma \to 1$ recovers the *undiscounted* results: $S_0, T_0, S_1, C \to N$ and $A_\gamma, B_\gamma \to A/N^2, B/N^2$, where $A, B$ are the non-discounted triangle sums.
(ii) As $N \to \infty$ with fixed $\gamma < 1$, $S_1 \to \gamma^2/(1-\gamma^2)$ and $C \to \gamma/(1-\gamma)$ while $N^2\gamma^{2N} \to 0$; thus $R_B(N, \gamma, \kappa) \to 0$. For typical RL regimes ($\gamma \gtrsim 0.95$, moderate $N$), $\kappa_\star(N, \gamma)$ is positive and large, so averaging still reduces bootstrap variance over a wide range of $\kappa$.
(iii) $R_\gamma(N)$ is horizon- and discount-agnostic in the sense of the bound $1 \le R_\gamma(N) < 4$; for $\gamma < 1$, its large-$N$ limit is 1. It is the principal driver of the overall variance reduction.

**On the equicorrelation assumption.** We assumed an equicorrelation (exchangeable) model for the reward noise and for the bootstrapped values: identical variances and a common pairwise correlation ($\rho$ and $\kappa$, respectively). This is a standard device that yields closed forms while capturing the empirically relevant regime of positively correlated temporal signals in RL trajectories.

The key conclusions above are *robust* to relaxing equicorrelation. Let $\Sigma$ be any covariance matrix for $(r_0, \ldots, r_{N-1})$ with nonnegative entries (i.e., nonnegative autocovariances). For any nonnegative weight vector $w$, the variance is $w^T\Sigma w$ and increases monotonically with each off-diagonal entry. Since the triangular weights have strictly smaller $\ell_2$ norm and smaller sum than the flat weights of the single $N$-step sum, the reward-side variance reduction persists under a wide range of stationary, positively correlated processes (including Toeplitz/lag-dependent models such as $\mathrm{Cov}(r_k, r_\ell) = \sigma^2\rho_{|k-\ell|}$ with $\rho_d \ge 0$). The specific constant 4 in the upper bound is tight for the exchangeable model; with general lag structure the same $[1, 4)$ bracket continues to hold under mild bounded-correlation conditions (e.g. $\sup_{k \ne \ell} \mathrm{Corr}(r_k, r_\ell) \le 1$), while the uncorrelated case ($\rho_d \equiv 0$) recovers the $[1, 3)$ limit.

For the bootstrap part, assuming a common correlation $\kappa$ across $\{Z_i\}$ is likewise a tractable approximation: the explicit ratio $R_B(N, \gamma, \kappa)$ is decreasing in $\kappa$, so weaker dependence only strengthens the variance reduction. More general lag-dependent models $\mathrm{Cov}(Z_i, Z_j) = \tau^2\kappa_{|i-j|}$ with $\kappa_d \ge 0$ lead to the same qualitative behavior (smaller weights and partial averaging reduce variance), with our equicorrelation formulas serving as convenient upper/lower benchmarks.

*When to be cautious.* If the process exhibits strong *negative* or oscillatory correlations (e.g. alternation effects), equicorrelation overstates the benefit of averaging; in such cases, replacing the common $\rho$ (or $\kappa$) by a small set of lag-specific parameters ($\rho_1, \rho_2, \ldots$) is safer. Empirically, one can estimate the sample autocovariance and plug it into $w^T\hat{\Sigma}w$ to verify the inequalities numerically.

# D    APPENDIX: VARIANCE REDUCTION FROM GRADIENT-LEVEL AVERAGING WITH A SHARED-WEIGHTS TRANSFORMER CRITIC

**Setup.**    Fix a trajectory position $t$. A Transformer critic with shared parameters $\psi$ outputs

$$Q_\psi^{(1)}, Q_\psi^{(2)}, \ldots, Q_\psi^{(n)},$$

where $Q_\psi^{(i)}$ predicts the $i$-step return for the same prefix $(s_t, a_t, \ldots, a_{t+i-1})$. Let $G^{(i)}$ denote the $i$-step target and define the per-horizon MSE

$$L_i(\psi) = \tfrac{1}{2}\big(Q_\psi^{(i)} - G^{(i)}\big)^2, \qquad \bar{L}(\psi) \triangleq \frac{1}{n}\sum_{i=1}^{n} L_i(\psi).$$

In implementation we *average their gradients* during backprop:

$$\nabla_\psi \bar{L}(\psi) \;=\; \frac{1}{n}\sum_{i=1}^{n} \nabla_\psi L_i(\psi) \;=\; \frac{1}{n}\sum_{i=1}^{n} \big(Q_\psi^{(i)} - G^{(i)}\big)\nabla_\psi Q_\psi^{(i)}.$$

**Local gradient factorization under weight sharing.**    Let $w$ be any scalar entry of $\psi$. By the chain rule,

$$g_i(w) \;\triangleq\; \frac{\partial L_i}{\partial w} = \underbrace{\big(Q_\psi^{(i)} - G^{(i)}\big)}_{\varepsilon^{(i)}} \underbrace{\frac{\partial Q_\psi^{(i)}}{\partial w}}_{\psi^{(i)}(w)}.$$

For linear modules (affine maps in attention/FFN), the Jacobian has the standard local form $\frac{\partial Q_\psi^{(i)}}{\partial w} = a^{(i)}\delta^{(i)}$ (input activation $\times$ upstream error). Because the same $w$ is *shared* across decoder positions, the sequence $\{g_i(w)\}_{i=1}^{n}$ are $n$ gradient contributions for the *same* parameter, drawn from adjacent positions of one forward pass, and are therefore generally *positively correlated*.

**A convenient covariance model (exchangeable/equicorrelated).**    For fixed $w$, we use the standard homoscedastic equicorrelation approximation (also common in mini-batch analyses):

$$\mathrm{Var}[g_i(w)] = \sigma_w^2, \qquad \mathrm{Cov}(g_i(w), g_j(w)) = \rho_w\,\sigma_w^2 \quad (i \neq j), \qquad \rho_w \in [0,1).$$

This captures the empirically relevant regime where adjacent horizons produce positively correlated gradients and yields tight, closed-form variance expressions.

**Lemma 3** (Variance of the averaged per-parameter update). *With the model above, the averaged update $\bar{g}(w) \triangleq \frac{1}{n}\sum_{i=1}^{n} g_i(w)$ satisfies*

$$\mathrm{Var}[\bar{g}(w)] = \frac{1}{n^2}\left(\sum_{i=1}^{n}\mathrm{Var}[g_i] + \sum_{i\neq j}\mathrm{Cov}(g_i, g_j)\right) = \sigma_w^2\,\frac{1 + (n-1)\rho_w}{n}.$$

*In particular, $\mathrm{Var}[\bar{g}(w)] < \sigma_w^2$ for any $\rho_w < 1$.*

**Corollary 3** (Effective batch size and asymptotics). *Define the* effective sample size $n_{\mathrm{eff}}(w) \triangleq \dfrac{n}{1 + (n-1)\rho_w}$. *Then $\mathrm{Var}[\bar{g}(w)] = \sigma_w^2/n_{\mathrm{eff}}(w)$ with $1 \le n_{\mathrm{eff}}(w) \le n$, strictly increasing in $n$, and $\lim_{n\to\infty}\mathrm{Var}[\bar{g}(w)] = \rho_w\,\sigma_w^2$ (the correlation-imposed variance floor).*

**Proposition 4** (Uniform horizon averaging is optimal under exchangeability). *Among all unbiased linear combinations $\sum_{i=1}^{n}\alpha_i g_i(w)$ with $\sum_i \alpha_i = 1$, the variance is minimized by the* uniform weights $\alpha_i = \frac{1}{n}$ whenever $\mathrm{Cov}(g_i, g_j)$ is exchangeable (same diagonal/off-diagonal).

*Proof.* For an exchangeable covariance $\Sigma_w = \sigma_w^2[(1 - \rho_w)I + \rho_w\mathbf{1}\mathbf{1}^T]$, $\mathrm{Var}(\sum_i \alpha_i g_i) = \alpha^T\Sigma_w\alpha$ is minimized under $\mathbf{1}^T\alpha = 1$ by $\alpha^\star = \frac{1}{n}\mathbf{1}$. $\qquad\square$

**Why $\rho_w \gtrsim 0$ is natural.**    Both multiplicative factors of $g_i(w)$ vary smoothly with $i$: (i) the targets $G^{(i)}$ share overlapping reward sums and a common bootstrapped tail; and (ii) the Jacobians $\partial Q_\psi^{(i)}/\partial w$ come from *adjacent* decoder positions of the same Transformer. This induces positive correlation among $\{g_i(w)\}$, putting us squarely in the regime of Lemma 3.

**Connection to target-side variance (discounted rewards and bootstrap).** Let $\gamma \in (0, 1]$ be the discount. Write the single $N$-step reward sum and its triangular average as

$$R_N(\gamma) = \sum_{k=0}^{N-1} \gamma^k r_k, \qquad \bar{R}_N(\gamma) = \frac{1}{N} \sum_{i=1}^{N} \sum_{k=0}^{i-1} \gamma^k r_k = \sum_{k=0}^{N-1} \underbrace{\frac{N-k}{N} \gamma^k}_{w_k} r_k.$$

Under the equicorrelated reward model (mean $\mu$, variance $\sigma^2$, pairwise corr. $\rho \geq 0$),

$$\mathrm{Var}[R_N(\gamma)] = \sigma^2 \big[ S_0 + \rho\,(T_0^2 - S_0) \big], \qquad \mathrm{Var}[\bar{R}_N(\gamma)] = \sigma^2 [A_\gamma + \rho\,B_\gamma],$$

with

$$S_0 = \sum_{k=0}^{N-1} \gamma^{2k} = \frac{1-\gamma^{2N}}{1-\gamma^2}, \quad T_0 = \sum_{k=0}^{N-1} \gamma^k = \frac{1-\gamma^N}{1-\gamma}, \quad A_\gamma = \sum_{k=0}^{N-1} w_k^2, \quad B_\gamma = \Big( \sum_{k=0}^{N-1} w_k \Big)^2 - A_\gamma.$$

The reward-side ratio

$$R_\gamma(N) \triangleq \frac{\mathrm{Var}[R_N(\gamma)]}{\mathrm{Var}[\bar{R}_N(\gamma)]} = \frac{S_0 + \rho\,(T_0^2 - S_0)}{A_\gamma + \rho B_\gamma}$$

satisfies the *uniform bound* $1 \leq R_\gamma(N) < 4$ for all $N \geq 1$, $\rho \geq 0$, $\gamma \in (0, 1]$, and $R_\gamma(N) \nearrow 4$ as $N \to \infty$.

For the bootstrapped values $Z_i = V_{\phi_{\mathrm{tar}}}(s_{t+i})$ with $\mathrm{Var}(Z_i) = \tau^2$ and $\mathrm{Cov}(Z_i, Z_j) = \kappa \tau^2$ ($i \neq j$, $\kappa \in [0, 1]$), define

$$B_N(\gamma) = \gamma^N Z_N, \qquad \bar{B}_N(\gamma) = \frac{1}{N} \sum_{i=1}^{N} \gamma^i Z_i,$$

and let $S_1 = \sum_{i=1}^{N} \gamma^{2i}$, $C = \sum_{i=1}^{N} \gamma^i$. Then

$$\mathrm{Var}[\bar{B}_N(\gamma)] = \frac{\tau^2}{N^2} [S_1 + \kappa(C^2 - S_1)], \qquad R_B(N, \gamma, \kappa) \triangleq \frac{\mathrm{Var}[B_N(\gamma)]}{\mathrm{Var}[\bar{B}_N(\gamma)]} = \frac{N^2 \gamma^{2N}}{S_1 + \kappa(C^2 - S_1)}.$$

$R_B$ is decreasing in $\kappa$ and obeys

$$\frac{N^2 \gamma^{2N}}{C^2} \leq R_B(N, \gamma, \kappa) \leq \frac{N^2 \gamma^{2N}}{S_1}.$$

In particular, averaging the bootstrap part reduces variance whenever $\kappa \leq \kappa_\star(N, \gamma) \triangleq \frac{N^2 \gamma^{2N} - S_1}{C^2 - S_1}$.

**Theorem 5** (Main: gradient averaging reduces update variance; compounded by target-side smoothing). *Let $w$ be any scalar parameter of the shared-weights Transformer critic and suppose $\{g_i(w)\}_{i=1}^n$ are homoscedastic and equicorrelated with $\rho_w < 1$. Then*

$$\mathrm{Var}\left[ \frac{\partial \bar{L}}{\partial w} \right] = \sigma_w^2 \frac{1 + (n-1)\rho_w}{n} < \sigma_w^2 = \mathrm{Var}\left[ \frac{\partial L_j}{\partial w} \right], \quad \forall j \in \{1, \dots, n\}.$$

*Moreover, writing $g_i(w) = \varepsilon^{(i)} \psi^{(i)}(w)$ and (mildly) assuming $\{\varepsilon^{(i)}\}$ and $\{\psi^{(i)}(w)\}$ are independent across $i$ with bounded second moments, there exist constants $a_w, b_w \geq 0$ (depending only on $\psi$) such that*

$$\mathrm{Var}[g_i(w)] \leq a_w \mathrm{Var}[G^{(i)}] + b_w.$$

*Consequently, replacing a single horizon by the triangularly averaged target across horizons 1:N reduces the reward-side variance by at least a factor $R_\gamma(N)^{-1} \in (1/4, 1]$, and (when $\kappa \leq \kappa_\star$) also reduces the bootstrap-side variance by a factor $R_B(N, \gamma, \kappa)^{-1}$. Thus, in addition to the* across-horizon gradient averaging *gain $\frac{1+(n-1)\rho_w}{n}$, the per-horizon variance term $\sigma_w^2$ itself decreases with $N$ via target-side smoothing, yielding a compounded reduction.*

**Practical notes.** (i) The gradient-level algebra is agnostic to discount $\gamma$; only the target-side constants $(S_0, T_0, A_\gamma, B_\gamma)$ and $(S_1, C)$ change with $\gamma$. (ii) Under exchangeability, uniform averaging across horizons is *variance-optimal* (Prop. 4); no learned horizon-weights are needed for variance reasons. (iii) As $n$ grows, the residual variance floor is $\rho_w \sigma_w^2$ (Cor. 3); lower temporal correlation between horizon-gradients directly improves this floor. (iv) If horizon-gradients are not perfectly exchangeable, the bound $\mathrm{Var}[\bar{g}(w)] \leq \frac{\bar{\sigma}_w^2}{n}\big(1 + (n-1)\bar{\rho}_w\big)$ still holds whenever $\mathrm{Var}[g_i] \leq \bar{\sigma}_w^2$ and $\mathrm{Corr}(g_i, g_j) \leq \bar{\rho}_w$ for all $i \neq j$.

# E APPENDIX: CONNECTION TO MULTI-STEP TD THEORY

**Motivation: $Q$ mixes environment and policy.** In an MDP $\mathcal{M} = (\mathcal{S}, \mathcal{A}, P, r, \gamma)$ and policy $\pi$,

$$Q^\pi(s,a) = \underbrace{r(s,a)}_{\text{env dynamics}} + \gamma \underbrace{\mathbb{E}_{s' \sim P(\cdot|s,a)}[V^\pi(s')]}_{\text{env dynamics + policy via } V^\pi} . \qquad V^\pi(s) = \mathbb{E}_{a \sim \pi(\cdot|s)}[Q^\pi(s,a)].$$

The one-step reward term is fully environment-determined (via $(P, r)$), while the bootstrap term depends on both the environment and the continuation policy through $V^\pi$. Our construction makes this separation explicit by defining critics that condition on a longer *realized* action prefix, thereby pushing all policy-dependence to a single boundary bootstrap term discounted by $\gamma^i$.

## E.1 PREFIX-CONDITIONED (EXTENDED-ACTION) $Q$

Fix a horizon $i \geq 1$ and write an action prefix as

$$\mathbf{a}_{0:i-1} \doteq (a_t, a_{t+1}, \ldots, a_{t+i-1}) \in \mathcal{A}^i, \qquad a_{t:t+i-1} = \mathbf{a}_{0:i-1}.$$

Define the **prefix-conditioned action-value** under continuation policy $\pi$ by

$$Q_i^\pi(s_t, \mathbf{a}_{0:i-1}) \doteq \mathbb{E}\left[\sum_{j=0}^{i-1} \gamma^j r_{t+j} + \gamma^i V^\pi(s_{t+i}) \,\middle|\, s_t, \, a_{t:t+i-1} = \mathbf{a}_{0:i-1}\right]. \qquad (13)$$

Conditioned on $(s_t, \mathbf{a}_{0:i-1})$, the random variables $(r_{t:t+i-1}, s_{t+i})$ are distributed *only* according to the environment dynamics induced by executing the prefix open-loop; the policy enters solely through the single bootstrap term $V^\pi(s_{t+i})$.

**Environment/policy decomposition.** Let the environment-determined prefix return be

$$R_i^{\text{env}}(s_t, \mathbf{a}_{0:i-1}) \doteq \mathbb{E}\left[\sum_{j=0}^{i-1} \gamma^j r_{t+j} \,\middle|\, s_t, \, a_{t:t+i-1} = \mathbf{a}_{0:i-1}\right].$$

Then equation 13 becomes

$$Q_i^\pi(s_t, \mathbf{a}_{0:i-1}) = R_i^{\text{env}}(s_t, \mathbf{a}_{0:i-1}) + \gamma^i \, \mathbb{E}[V^\pi(s_{t+i}) \,|\, s_t, \, a_{t:t+i-1} = \mathbf{a}_{0:i-1}]. \qquad (14)$$

As $i$ grows, more of the return is captured by realized (environment-defined) rewards, while the policy-dependent component is confined to the boundary and attenuated by $\gamma^i$.

## E.2 BELLMAN VIEW VIA AN AUGMENTED MDP (EXTENDED ACTION SPACE)

For fixed $i$, define an augmented MDP $\mathcal{M}_i = (\mathcal{S}, \mathcal{A}^i, P_i, r_i, \gamma^i)$ where an extended action $\mathbf{a}_{0:i-1}$ is executed open-loop for $i$ steps:

$$P_i(s'|s, \mathbf{a}_{0:i-1}) \doteq \Pr(s_{t+i} = s' \mid s_t = s, \, a_{t:t+i-1} = \mathbf{a}_{0:i-1}), \qquad (15)$$

$$r_i(s, \mathbf{a}_{0:i-1}) \doteq \mathbb{E}\left[\sum_{j=0}^{i-1} \gamma^j r_{t+j} \,\middle|\, s_t = s, \, a_{t:t+i-1} = \mathbf{a}_{0:i-1}\right]. \qquad (16)$$

Then equation 13 is exactly the one-step Bellman equation in $\mathcal{M}_i$:

$$Q_i^\pi(s, \mathbf{a}) = r_i(s, \mathbf{a}) + \gamma^i \, \mathbb{E}_{s' \sim P_i(\cdot|s,\mathbf{a})}[V^\pi(s')]. \qquad (17)$$

Hence, learning $Q_i^\pi$ from replayed prefixes is simply multi-step TD on an MDP whose actions are length-$i$ open-loop prefixes.

## E.3 WHY NO IMPORTANCE SAMPLING IS NEEDED (BY CONSTRUCTION)

Let $\mu$ be any behavior policy producing replay. Given a sampled window/prefix $(s_t, \mathbf{a}_{0:i-1}, r_{t:t+i-1}, s_{t+i})$, define the target (as in Eq.(5) of the main text, using a target value network $V_\phi$)

$$G^{(i)}(s_t, \mathbf{a}_{0:i-1}) \doteq \sum_{j=0}^{i-1} \gamma^j r_{t+j} + \gamma^i V_\phi(s_{t+i}). \qquad (18)$$

**Proposition 6** (Conditional unbiasedness; no dependence on $\mu$)**.** *For any fixed* $(s_t, \mathbf{a}_{0:i-1})$,

$$\mathbb{E}\left[\sum_{j=0}^{i-1} \gamma^j r_{t+j} + \gamma^i V^\pi(s_{t+i}) \,\middle|\, s_t,\ a_{t:t+i-1} = \mathbf{a}_{0:i-1}\right] = Q_i^\pi(s_t, \mathbf{a}_{0:i-1}).$$

*In particular, this conditional expectation is determined by* $(P, r)$ *and* $\pi$ *(only through* $V^\pi$*), and is independent of the behavior policy* $\mu$ *that generated the prefix.*

*Proof sketch.* Conditioning on $(s_t, a_{t:t+i-1} = \mathbf{a}_{0:i-1})$ fixes the executed actions. The induced distribution of $(r_{t:t+i-1}, s_{t+i})$ is therefore the environment distribution under open-loop execution of the prefix from $s_t$; $\mu$ is irrelevant after conditioning. Taking expectations yields equation 13. $\square$

**Contrast with standard off-policy $i$-step evaluation of $Q^\pi(s_t, a_t)$.** If one instead targets the classical $Q^\pi(s_t, a_t)$ in the original MDP, then intermediate actions $a_{t+1:t+i-1}$ must be integrated under $\pi$, while replay provides them under $\mu$, creating a distribution mismatch that typically requires per-decision importance sampling. Here, the intermediate actions are part of the *conditioning variable* (we learn $Q_i^\pi(s_t, \mathbf{a}_{0:i-1})$), so there is no such mismatch to correct.

### E.4 Relation to the standard one-step $Q^\pi$

This definition is consistent with the classical action-value:

$$Q_1^\pi(s, a) = Q^\pi(s, a).$$

For $i > 1$, the original $Q^\pi(s, a)$ is recovered by marginalizing the future actions under $\pi$:

$$Q^\pi(s, a) = \mathbb{E}_{\substack{a_{1:i-1} \sim \pi \\ s_{1:i} \sim P}}\left[Q_i^\pi\big(s, (a, a_1, \ldots, a_{i-1})\big)\right]. \tag{19}$$

Thus $Q_i^\pi$ can be viewed as a refinement that conditions on a longer realized prefix, moving more of the return into environment-realized rewards and leaving only a discounted boundary bootstrap term to carry policy dependence.

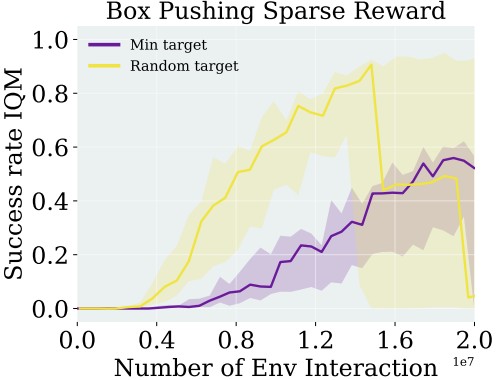 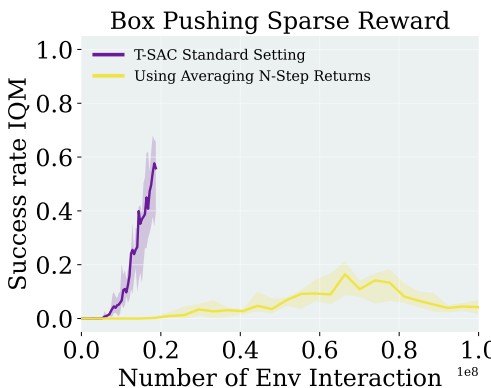

**(a)** Effect of target-value construction under sparse rewards. Randomly selecting one of two double-$Q$ targets leads to high-variance updates and occasional collapse, whereas the conservative minimum-based target yields stable learning curve with the hard–copy critic.

**(b)** Naively averaging $N$-step targets across horizons destabilizes learning and can erase progress, confirming the need for gradient–level averaging.

**Figure 9: Meta-World Box Pushing (Sparse Reward).** Ablations on (a) target-value construction and (b) return-propagation schemes for T–SAC.

## F APPENDIX: ADDITIONAL EXPERIMENTAL RESULTS

**Post-figure summary.** Figure 9a shows that the instability originally observed on Box–Pushing–Sparse is explained by the high–variance target estimator: with the conservative minimum–based target, hard–copy T–SAC is stable and reaches the best success rates. Figure 9b further illustrates that naive $N$-step target averaging can derail optimization, motivating our choice of gradient–averaged multi–horizon losses. **Seeds: 4. Results under IQM with 95% confidence intervals.**

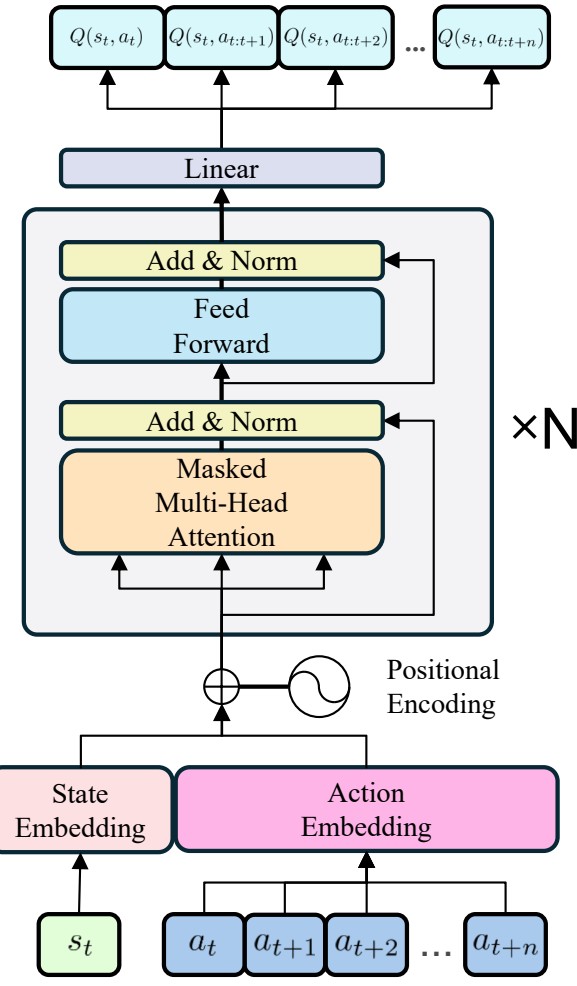

**Figure 10:** T-SAC Critic Detailed Structure: a causal Transformer over short state–action segments. Given $(s_t, a_t, \ldots, a_{t+n})$, the network produces $n$ scalar outputs $\{Q_\psi(s_t, a_t, \ldots, a_{t+i})\}_{i=1}^n$. Colors and block styling follow the Transformer diagram conventions of Vaswani et al. (2017).

## G  APPENDIX: TRANSFORMER CRITIC DETAILED STRUCTURE

**Implementation details.** We follow the TOP–ERL–style Transformer critic design adopted in this work (see Li et al. (2024a) for the schematic), i.e., a masked multi–head self–attention stack with positional encodings and residual Add&Norm blocks; the critic ingests $(s_t, a_t, \ldots, a_{t+n})$ and jointly predicts all $1 \ldots n$ step returns. State and action tokens use *separate* one–layer linear embeddings (no bias), consistent with our training objective that conditions on realized action prefixes; the output head is a linear map without bias that emits one scalar per decoder position. No dropout is used anywhere in the critic. The causal mask ensures each position $i$ only attends to $\leq i$ tokens, aligning the network outputs with the $i$-step targets used for learning.

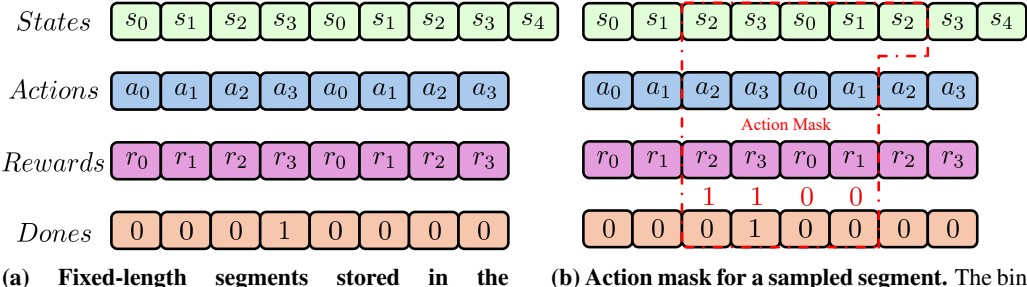

**(a) Fixed-length segments stored in the replay buffer.** Each sample is a window $(s_{t:t+L}, a_{t:t+L-1}, r_{t:t+L-1}, d_{t:t+L-1})$. If a terminal $d_\tau = 1$ appears before the window is full, we immediately continue saving from the start of the next trajectory until the segment length $L$ is reached.

**(b) Action mask for a sampled segment.** The binary mask $m$ marks valid action positions: $m_\tau = 1$ for steps before or at the first terminal in the window and $m_\tau = 0$ after any $d_\tau = 1$. Entries that occur after a terminal (including those filled from a new trajectory) are masked out so losses/targets and attention never cross episode boundaries.

Figure 11: T-SAC segment construction and masking.

# H APPENDIX: SEGMENT CONSTRUCTION AND THEORETICAL BENEFITS OF CRITIC CHUNKING

## H.1 TRAJECTORIES SAVED IN REPLAY BUFFER AND ACTION MASK DESIGN

To efficiently process sequence data and maximize hardware utilization, T-SAC stores experience in the replay buffer as continuous, fixed-length segments rather than isolated single transitions. As illustrated in Figure 11, a typical stored segment consists of a sequence of states, actions, rewards, and done flags: $(s_{t:t+L}, a_{t:t+L-1}, r_{t:t+L-1}, d_{t:t+L-1})$. In environments with variable-length episodes, a trajectory may terminate before the fixed segment capacity $L$ is reached. Rather than padding the remainder of the buffer with zeros—which wastes memory and compute—we immediately append the initial transitions of the subsequent episode to the same segment until the length $L$ is satisfied. This strategy ensures constant tensor shapes, heavily optimizing batch processing for the Transformer critic.

However, packing multiple disjoint trajectories into a single contiguous segment introduces a mathematical risk: the critic might incorrectly bootstrap values or attend to states across episode boundaries. To strictly prevent this, we implement a straightforward binary action mask, $m$.

As detailed in Figure 11, the mask identifies the first terminal flag within a sampled window. We set $m_\tau = 1$ for all timesteps up to and including the first terminal transition, and strictly enforce $m_\tau = 0$ for all subsequent steps within that specific window. This action mask is applied when constructing the variable-horizon $N$-step targets, the mask zeroes out the TD error for any horizon that attempts to bootstrap across the boundary.

By doing so, any entries occurring after a terminal state are entirely ignored in the loss formulation. This masking acts as an implementation detail that resolves the mismatch between variable-length environments and fixed-length sequence models. It fully preserves the Markov properties of the MDP by preventing boundary leakage without degrading performance.

## H.2 THE BENEFIT OF CHUNKING ON THE CRITIC SIDE

We analyze chunking under sparse rewards and state coverage.

**Setup.** Consider a fixed segment of $N$ transitions $(s_1, a_1, r_1, \ldots, s_{N+1})$. A *chunked* sample is obtained by: (i) drawing a start index $p$ uniformly from $\{1, \ldots, N\}$; (ii) drawing a window length $L$ uniformly from the integers $\{\ell_{\min}, \ldots, \ell_{\max}\}$ (with $\ell_{\min} \geq 1$). The window is truncated at the episode boundary, yielding an effective sequence length

$$h = \min\{L, N - p + 1\}.$$

For the *fixed* start state $s_p$, the Transformer critic predicts the $i$-step returns for every horizon $i \in \{1, \ldots, h\}$. The multi-step target for horizon $i$ is:

$$G^{(i)} = \sum_{j=0}^{i-1} \gamma^j r_{p+j} + \gamma^i V_\phi(s_{p+i}). \tag{20}$$

Unlike standard multi-step methods that compute a single return, T-SAC computes $h$ distinct targets—each with a different horizon and bootstrap state—from a single sampled start state $s_p$.

**Connection to state coverage (selection) probability.** The probability that a given transition $j$ is *covered* by the sampled window (i.e., $j \in [p, \ p+h-1]$) is

$$\Pr(j \text{ covered}) = \frac{1}{N\,\Delta} \sum_{p=1}^{j} \left[\ell_{\max} - \max\{\ell_{\min}, \ j-p+1\} + 1\right]_+, \tag{21}$$

where $\Delta = \ell_{\max} - \ell_{\min} + 1$ and $[\cdot]_+ = \max\{\cdot, 0\}$. For the common case $\ell_{\min} = 1$ and writing $m = \ell_{\max}$, this simplifies to

$$\Pr(j \text{ covered}) = \begin{cases} \dfrac{1}{N}\left(j - \dfrac{j(j-1)}{2m}\right), & 1 \le j \le m, \\[2ex] \dfrac{m+1}{2N}, & m < j \le N, \end{cases} \tag{22}$$

i.e., a ramp near the start followed by a flat plateau. This higher coverage (vs. 1-step sampling) provides a denser learning signal across the trajectory.

**Sparse rewards: how far does a single reward propagate?** Assume only the terminal transition $N$ carries a non-zero reward (the sparse-reward setting). For a sampled window starting at $p$ with effective length $h$, the targets $G^{(1)}, \ldots, G^{(h)}$ are computed. The terminal reward $r_N$ is included in a target if and only if the window reaches the end of the episode (i.e., $p+h-1 = N$). Crucially, it will be included in *exactly one* horizon's target (specifically, $i = N - p + 1$).

Therefore, the expected number of reward-bearing updates per sampled window is exactly the probability that the sampled window covers the terminal transition:

$$\mathbb{E}\big[\#\text{updates including terminal reward}\big] = \Pr(N \text{ covered}). \tag{23}$$

Assuming $N \ge \ell_{\max}$, this is simply the plateau value of the coverage probability:

$$\mathbb{E}\big[\#\text{updates including terminal reward}\big] = \frac{1}{N\,\Delta} \sum_{L=\ell_{\min}}^{\ell_{\max}} L = \frac{\mathbb{E}[L]}{N}. \tag{24}$$

In the standard setting with $\ell_{\min} = 1$:

$$\mathbb{E}\big[\#\text{updates including terminal reward}\big] = \frac{\ell_{\max} + 1}{2N}, \tag{25}$$

representing an $\mathbb{E}[L]$-fold amplification (e.g., an $8.5$-fold increase for $\ell_{\max} = 16$) over uniform 1-step TD, which touches the terminal reward only when $p = N$ (a $1/N$ probability).

**Takeaways.** Chunking yields two critical critic-side benefits: (i) *Denser state coverage:* Instead of updating a single state-action value per sample, each sampled window generates $\mathbb{E}[L]$ distinct multi-horizon updates starting from $s_p$. This dramatically increases the coverage probability of states across the trajectory, accelerating value propagation.
(ii) *Sparse-reward propagation:* When only the last transition is rewarded, chunking increases the proportion of updates that incorporate the true reward by a factor of $\mathbb{E}[L]$. By naturally wrapping the terminal reward into the exact-horizon target, this mechanism explains why T-SAC excels in long-horizon, sparse-reward settings without suffering from extreme off-policy variance.

# I  APPENDIX: COMPUTATIONAL COSTS AND SAMPLE EFFICIENCY

Training time is reported for 1M environment steps (UTD= 1), unless otherwise stated. All benchmarks were run on an NVIDIA A100 (40 GB) GPU and an Intel Xeon Platinum 8368 CPU. Network architectures are configured as follows:

**Figure 12:** Training time is reported for 1M environment steps (UTD= 1), unless otherwise stated. All benchmarks were run on an NVIDIA A100 (40 GB) GPU and an Intel Xeon Platinum 8368 CPU.

- **T-SAC (ours):** MLP policy with two 256-unit hidden layers; Transformer critic with 2 layers × 256 units.

- **GRU/LSTM:** Same policy; 2-layer RNN critic (256 units).

- **SAC and CrossQ:** Default configurations.

**Table 1:** Sample efficiency on long-horizon benchmarks, measured as the number of environment steps (in millions) required to reach a fixed performance threshold on each task. Thresholds are defined as 90% of SAC's final return on Box-Pushing (dense) and ML1, and 90% of T-SAC's final return on Box-Pushing (sparse). Lower is better. Values are means over seeds.

| Task | SAC | CrossQ | GTrXL policy | TOP-ERL | T-SAC (ours) |
|---|---|---|---|---|---|
| Box-Pushing (dense) | 15M | 10M | 20M | **2M** | 4M |
| Box-Pushing (sparse) | N.A. | N.A. | N.A. | **4M** | 17M |
| ML1 | 4M | N.A. | N.A. | 4M | **1M** |

**Table 2:** Effect of minimum and maximum sequence length on T-SAC performance and wall-clock training time on *Box-Pushing (dense)*. All runs use the same number of environment steps (1M) for the standard setting.

| min_length | max_length | Return (mean $\pm$ s.e.) | Wall-clock time | Peak GPU Mem (GB) |
|---|---|---|---|---|
| 1 | 1 | $-78.45 \pm 6.89$ | 2h35m03s | 2.37 |
| 4 | 4 | $-66.63 \pm 3.09$ | 2h39m23s | 2.37 |
| 1 | 4 | $-74.80 \pm 7.69$ | 2h38m56s | 2.37 |
| 1 | 16 | $\mathbf{-65.05 \pm 0.20}$ | **3h06m58s** | 2.37 |

**Table 3:** Computational cost comparison for different methods on *Box-Pushing (dense)* for a fixed number of environment steps and matched (or explicitly stated) update-to-data (UTD) ratios.

| Method | Params (M) | UTD | Wall-clock time (hours) |
|---|---|---|---|
| SAC | 0.2 | 1 | 1.47 |
| CrossQ | 10 | 1 | 2.44 |
| T-SAC (ours) | 3.3 | 1/4 | 0.77 |
| TOP-ERL | 3.3 | 1/10 | 0.35 |

**Table 4:** Performance at fixed data budgets on long-horizon tasks. Entries are success rate ± standard error over seeds after a given portion of environment steps. All methods are trained with the same number of transitions.

| Task | Method | 25% | 50% | 100% |
|---|---|---|---|---|
| Box-Pushing (dense) (max 20M steps) | SAC | $1\% \pm 1\%$ | $5\% \pm 1\%$ | $12\% \pm 3\%$ |
| | CrossQ | $3\% \pm 10\%$ | $10\% \pm 20\%$ | n.a. |
| | GTrXL policy | $1\% \pm 1\%$ | $2\% \pm 1\%$ | $10\% \pm 1\%$ |
| | T-SAC (ours) | $\mathbf{20\% \pm 3\%}$ | $\mathbf{80\% \pm 3\%}$ | $\mathbf{96\% \pm 1\%}$ |
| Box-Pushing (sparse) (max 20M steps) | SAC | $0\% \pm 1\%$ | $1\% \pm 1\%$ | $1\% \pm 1\%$ |
| | CrossQ | $0\% \pm 0\%$ | $1\% \pm 1\%$ | n.a. |
| | GTrXL policy | $0\% \pm 1\%$ | $1\% \pm 1\%$ | $1\% \pm 1\%$ |
| | T-SAC (ours) | $\mathbf{2\% \pm 1\%}$ | $\mathbf{15\% \pm 10\%}$ | $\mathbf{60\% \pm 25\%}$ |
| ML1 (max 5M steps) | SAC | $53\% \pm 7\%$ | $60\% \pm 10\%$ | $68\% \pm 10\%$ |
| | CrossQ | $50\% \pm 5\%$ | $50\% \pm 5\%$ | $50\% \pm 5\%$ |
| | GTrXL policy | $18\% \pm 5\%$ | $35\% \pm 5\%$ | $53\% \pm 13\%$ |
| | T-SAC (ours) | $\mathbf{68\% \pm 12\%}$ | $\mathbf{78\% \pm 10\%}$ | $\mathbf{86\% \pm 5\%}$ |

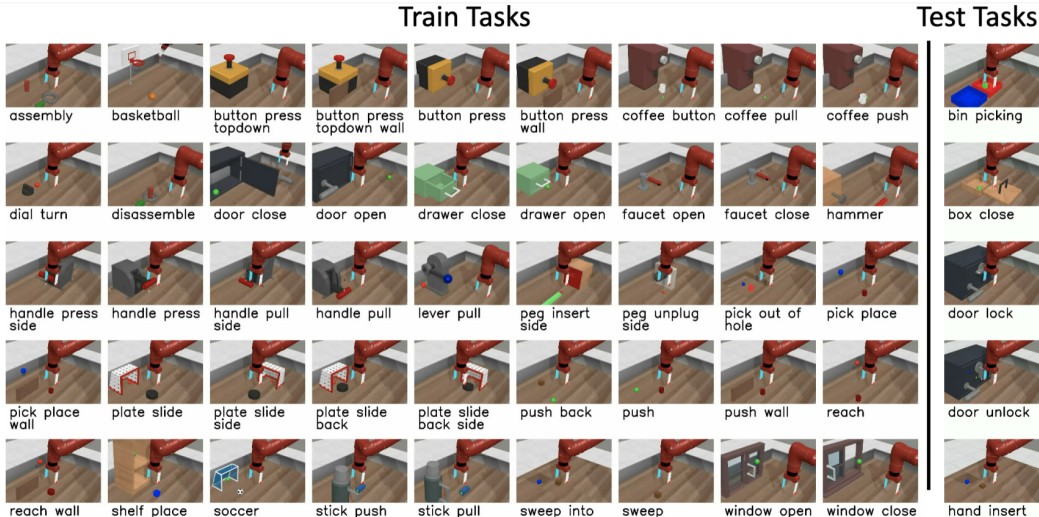

**Figure 13:** Meta-World tasks (Yu et al., 2020).

## J  EXPERIMENT DESCRIPTION

### J.1  META-WORLD ML1

Meta-World (Yu et al., 2020) is an open-source simulated benchmark for meta-reinforcement learning and multi-task learning in robotic manipulation. It comprises 50 distinct tasks spanning skills such as grasping, pushing, and object placement, each posing different perception–control challenges. By covering a broader skill spectrum than narrowly scoped benchmarks, Meta-World is well-suited for evaluating algorithms that aim to generalize across diverse behaviors. Figure 13 enumerates all 50 tasks and illustrates their variety and difficulty.

**Success criterion.** To better approximate real-world deployment, we adopt a stringent evaluation rule: an episode is counted as successful only if the environment's success condition is satisfied at the *final* timestep; intermediate achievements do not count toward success.

## J.2 BOX PUSHING

**Setup.** A 7-DoF Franka Emika Panda arm with a rod pushes a box on a table to a target pose. At episode start, initial and target box poses are sampled with a minimum $0.2\,\mathrm{m}$ separation:

$$x_i \in [0.3, 0.6], \quad y_i \in [-0.4, 0.4], \quad \theta \in [0, 2\pi].$$

Success (for evaluation) is position error $\leq 0.05\,\mathrm{m}$ and orientation error $\leq 0.5\,\mathrm{rad}$.

**Observations & Actions.** Observations: robot joint positions/velocities $(\boldsymbol{q}, \dot{\boldsymbol{q}})$, box position/orientation $(\boldsymbol{p}, \boldsymbol{r})$, and target $(\boldsymbol{p}_{\text{target}}, \boldsymbol{r}_{\text{target}})$. Actions: joint torques $\boldsymbol{a}_t \in \mathbb{R}^7$.

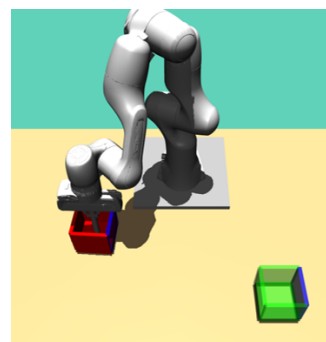

**Termination.** Fixed horizon $T = 100$ steps; no early termination.

**Figure 14:** Box Pushing task (Otto et al.).

**Dense reward.** At each step,

$$R_{\text{total}} = -R_{\text{rod}} - 0.02\,\tau_t - \text{err}(\boldsymbol{q}, \dot{\boldsymbol{q}}) - 350\,R_{\text{position}} - 200\,R_{\text{rotation}}.$$

Subterms are

$$R_{\text{rod}} = \text{clip}(\|\boldsymbol{p} - \boldsymbol{h}_{\text{pos}}\|,\, 0.05,\, 10) + \text{clip}\left(\tfrac{2}{\pi}\arccos|\boldsymbol{h}_{\text{rot}}\cdot\boldsymbol{h}_0|,\, 0.25,\, 2\right), \tag{26}$$

$$\tau_t = \sum_{i=1}^{7}(a_t^i)^2, \tag{27}$$

$$\text{err}(\boldsymbol{q}, \dot{\boldsymbol{q}}) = \sum_{i:\,|q_i|>|q_i^b|}\left(|q_i| - |q_i^b|\right) + \sum_{j:\,|\dot{q}_j|>|\dot{q}_j^b|}\left(|\dot{q}_j| - |\dot{q}_j^b|\right), \tag{28}$$

$$R_{\text{position}} = \|\boldsymbol{p} - \boldsymbol{p}_{\text{target}}\|, \tag{29}$$

$$R_{\text{rotation}} = \tfrac{1}{\pi}\arccos|\boldsymbol{r}\cdot\boldsymbol{r}_{\text{target}}|. \tag{30}$$

Here, $\boldsymbol{h}_{\text{pos}}$ is the rod tip position, and $\boldsymbol{h}_{\text{rot}}, \boldsymbol{h}_0$ are rod orientations (quaternions).

**Sparse reward.** Only the task terms are applied at the final step:

$$R_{\text{total}} = \begin{cases} -R_{\text{rod}} - 0.02\,\tau_t - \text{err}(\boldsymbol{q}, \dot{\boldsymbol{q}}), & t < T, \\ -R_{\text{rod}} - 0.02\,\tau_t - \text{err}(\boldsymbol{q}, \dot{\boldsymbol{q}}) - 350\,R_{\text{position}} - 200\,R_{\text{rotation}}, & t = T. \end{cases} \tag{31}$$

## J.3 GYMNASIUM MUJOCO

We evaluate on the Gymnasium MuJoCo **v4** suite—`Ant-v4`, `HalfCheetah-v4`, `Hopper-v4`, `Walker2d-v4`, and `HumanoidStandup-v4` (Fig. 15). We use the default observation and action spaces and the native v4 reward shaping and termination rules (no reward normalization). Performance is reported as undiscounted episode return. Unless noted otherwise, evaluation uses the deterministic policy over 152 episodes and aggregates results across multiple random seeds using the IQM with 95% bootstrap confidence intervals; full hyperparameters and seeds are provided in App. L.

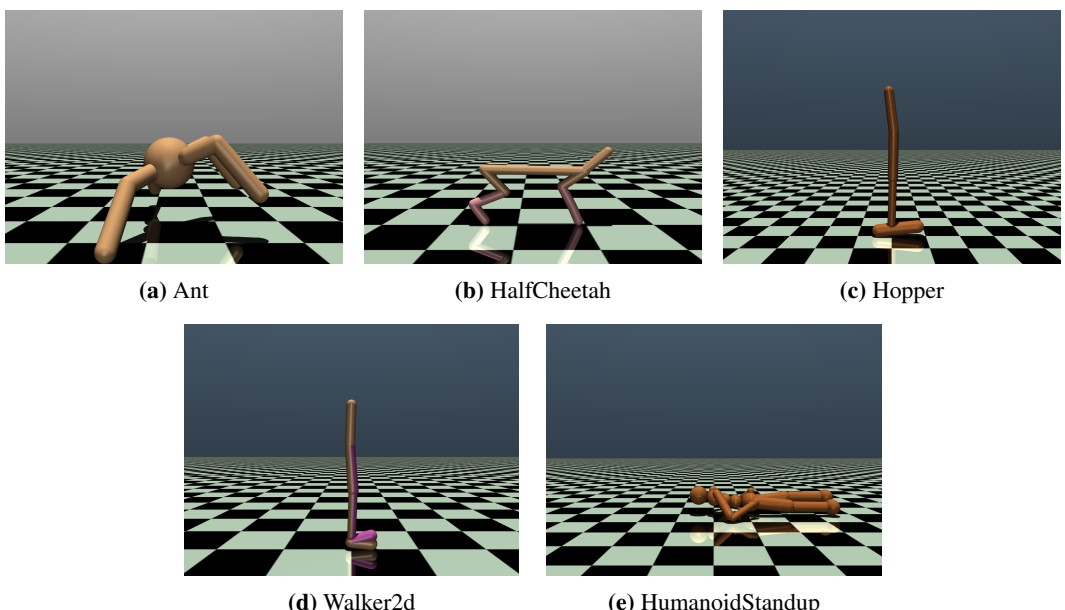

**(a)** Ant       **(b)** HalfCheetah       **(c)** Hopper

**(d)** Walker2d       **(e)** HumanoidStandup

**Figure 15:** MuJoCo (Towers et al., 2024) tasks used in our experiments.

## K  APPENDIX: ALGORITHM IMPLEMENTATIONS

**PPO**  Proximal Policy Optimization (PPO) (Schulman et al., 2017) is an on–policy, step–based method that constrains policy updates to remain close to the behavior policy. Two variants are common: *PPO–Penalty* (KL regularization) and *PPO–Clip* (clipped surrogate). We evaluate PPO–Clip given its prevalence and robustness, following the reference implementation in Raffin et al. (2021). **Seeds: 20.**

**SAC**  Soft Actor–Critic (SAC) (Haarnoja et al., 2018a;b) is an off–policy actor–critic with twin Q–networks to mitigate overestimation and an entropy term to encourage exploration. We use the Bhatt et al. (2019) implementation, which includes SAC. **Seeds: 20 (Meta–World ML1), 5 (Gym MuJoCo).**

**TD3**  Twin Delayed DDPG (TD3) (Fujimoto et al., 2018) addresses overestimation and instability via (i) clipped double Q–learning, (ii) delayed policy updates, and (iii) target policy smoothing. Our TD3 follows standard practice adapted from Raffin et al. (2021), including Polyak averaging and action noise for exploration. **Seeds: 5.**

**GTrXL**  Gated Transformer–XL (GTrXL) (Parisotto et al., 2020) stabilizes Transformer training for partially observable control. We build on the PPO + GTrXL implementation from Liang et al. (2018) and add minibatch advantage normalization plus a state–independent log–standard–deviation head following Huang et al. (2022). **Seeds: 4.**

**gSDE**  Generalized State–Dependent Exploration (gSDE) (Raffin et al., 2022; Rückstieß et al., 2008; Rückstiess et al., 2010) replaces i.i.d. Gaussian action noise with state–dependent, temporally smooth exploration. Concretely, disturbances are generated as $\epsilon_t = \Theta s$, where $s$ is the last hidden layer's activation and $\Theta$ is resampled from a Gaussian every $n$ steps according to the SDE sampling frequency. We evaluate gSDE with PPO using the reference implementation of Raffin et al. (2022); for stability on some tasks we employ a linear schedule for the PPO clipping range. **Seeds: 20.**

**BBRL**  Black–Box Reinforcement Learning (BBRL) (Otto et al., 2023a;b) performs episodic, trajectory–level search by parameterizing policies with ProMPs (Paraschos et al., 2013). This handles sparse and non–Markovian rewards but can reduce sample efficiency. We consider both di-

agonal–covariance (**BBRL–Std**) and full–covariance (**BBRL–Cov**) Gaussian policies, paired with ProDMP (Li et al., 2023). **Seeds: 20.**

**TCP** Temporally–Correlated Episodic RL (TCP) (Li et al., 2024b) augments episodic policy updates with step–level signals, narrowing the gap between episodic and step–based RL while retaining smooth, parameter–space exploration. **Seeds: 20.**

**TOP–ERL** Trajectory–Optimized Policy for Episodic RL (TOP–ERL) optimizes a distribution over motion–primitive parameters with (i) a KL–constrained trust region and (ii) a temporally structured covariance that induces smooth, correlated exploration across the episode. Our instantiation uses ProDMP (Li et al., 2023) as the trajectory generator; unless stated, we adopt an adaptive scale (entropy) schedule and per–dimension normalization of primitive parameters. **Seeds: 8.**

**CrossQ** CrossQ (Bhatt et al., 2019) is an off–policy SAC variant that removes target networks and applies BRN in the critic, enabling strong sample efficiency at an update–to–data ratio of $UTD = 1$. We follow the authors' reference implementation: a single batch–normalized critic (no target networks), default temperature tuning, and recommended hyperparameters unless stated otherwise. **Seeds: 4 (Meta–World ML1), 5 (Gym MuJoCo and Box–Pushing).** Training on Box–Pushing was capped at $10\,\mathrm{M}$ steps due to the experiment budget; by that point, wall–clock time exceeded $24\,\mathrm{h}$.

## L APPENDIX: HYPERPARAMETERS OF THE ALGORITHMS

**Baseline provenance.** For **BBRL**, **TCP**, **PPO**, **gSDE**, **GTrXL**, **TOP–ERL**, and **SAC** on Meta–World ML1, we report numbers from prior publications and/or official released runs/configurations under settings comparable to ours; we did not perform additional large–scale sweeps for these baselines in this paper (see citations in the main text and Appendix K).

**Methods tuned in this work.** We tuned **SAC** on Gym/FANCYGYM, the full **CrossQ** implementation, and **TD3**, including optimizer selection and hyperparameters (e.g., learning rates).

**Our tuning for T–SAC.** For **T–SAC**, we conducted a targeted grid search over Transformer–critic depth (number of attention layers), number of heads, dimensions per head, learning rates (policy/critic/$\alpha$), supervision–window settings (fixed vs. variable horizons; `min_length`, `step_length`), number of per–step target windows, and policy–side chunk length (for the compatibility study). Where appropriate, we initialized choices from publicly reported configurations: Transformer hyperparameter ranges from TOP–ERL (Li et al., 2024a), the entropy–temperature term from Celik et al. (2025), and the optimizer family from CrossQ (Bhatt et al., 2019). Final settings and search grids are listed in the appendix tables.

**Table 5:** Hyperparameters for the Meta-World experiments. Episode Length $T = 500$

| | PPO | gSDE | GTrXL | SAC | CrossQ | TCP | BBRL | TOP-ERL | T-SAC |
|---|---|---|---|---|---|---|---|---|---|
| number samples | 16000 | 16000 | 19000 | 1000 | 1 | 16 | 16 | 2 | 4 * 125 |
| GAE $\lambda$ | 0.95 | 0.95 | 0.95 | n.a. | n.a. | 0.95 | n.a. | n.a. | n.a. |
| discount factor | 0.99 | 0.99 | 0.99 | 0.99 | 0.99 | 1 | 1 | 1.0 | 0.99 |
| | | | | | | | | | |
| $\epsilon_\mu$ | n.a. | n.a. | n.a. | n.a. | n.a. | 0.005 | 0.005 | 0.005 | n.a. |
| $\epsilon_\Sigma$ | n.a. | n.a. | n.a. | n.a. | n.a. | 0.0005 | 0.0005 | 0.0005 | n.a. |
| trust region loss coef. | n.a. | n.a. | n.a. | n.a. | n.a. | 1 | 10 | 1.0 | n.a. |
| | | | | | | | | | |
| optimizer | adam | adam | adam | adam | adam | adam | adam | adam | adamw |
| epochs | 10 | 10 | 5 | 1000 | 1 | 50 | 100 | 15 | 20 |
| learning rate | 3e-4 | 1e-3 | 2e-4 | 3e-4 | 3e-4 | 3e-4 | 3e-4 | 1e-3 | 2.5e-4 |
| use critic | True | True | True | True | True | True | True | True | True |
| epochs critic | 10 | 10 | 5 | 1000 | 1 | 50 | 100 | 50 | 100 |
| learning rate critic | 3e-4 | 1e-3 | 2e-4 | 3e-4 | 3e-4 | 3e-4 | 3e-4 | 5e-5 | 2.5e-5 |
| number minibatches | 32 | n.a. | n.a | n.a. | n.a. | n.a. | n.a. | n.a. | n.a. |
| batch size | n.a. | 500 | 1024 | 256 | 256 | n.a. | n.a. | 256 | 512 |
| buffer size | n.a. | n.a. | n.a. | 1e6 | 1e6 | n.a. | n.a. | 3000 | 5000 * 125 |
| learning starts | 0 | 0 | n.a. | 10000 | 5000 | 0 | 0 | 2 | 200 |
| temperature warmup | 0 | 0 | 0 | 0 | 0 | 0 | 0 | 0 | 10000 |
| polyak_weight | n.a. | n.a. | n.a. | 5e-3 | 1.0 | n.a. | n.a. | 5e-3 | 5e-3 |
| SDE sampling frequency | n.a. | 4 | n.a. | n.a. | n.a. | n.a. | n.a. | n.a. | n.a. |
| entropy coefficient | 0 | 0 | 0 | auto | auto | 0 | 0 | n.a. | auto |
| | | | | | | | | | |
| normalized observations | True | True | False | False | False | True | False | False | False |
| normalized rewards | True | True | 0.05 | False | False | False | False | False | False |
| observation clip | 10.0 | n.a. | n.a. | n.a. | n.a. | n.a. | n.a. | n.a. | n.a. |
| reward clip | 10.0 | 10.0 | 10.0 | n.a. | n.a. | n.a. | n.a. | n.a. | n.a. |
| critic clip | 0.2 | lin_0.3 | 10.0 | n.a. | n.a. | n.a. | n.a. | n.a. | n.a. |
| importance ratio clip | 0.2 | lin_0.3 | 0.1 | n.a. | n.a. | n.a. | n.a. | n.a. | n.a. |
| | | | | | | | | | |
| hidden layers | [128, 128] | [128, 128] | n.a. | [256, 256] | [256, 256] | [128, 128] | [32, 32] | [ 128, 128] | [ 128, 128] |
| hidden layers critic | [128, 128] | [128, 128] | n.a. | [256, 256] | [2048, 2048] | [128, 128] | [32, 32] | n.a. | n.a. |
| hidden activation | tanh | tanh | relu | relu | relu | relu | relu | leaky_relu | leaky_relu |
| orthogonal initialization | Yes | No | xavier | fanin | fanin | Yes | Yes | Yes | fanin |
| initial std | 1.0 | 0.5 | 1.0 | 1.0 | 1.0 | 1.0 | 1.0 | 1.0 | 1.0 |
| number of heads | - | - | 4 | - | - | - | - | 8 | 4 |
| dims per head | - | - | 16 | - | - | - | - | 16 | 32 |
| number of attention layers | - | - | 4 | - | - | - | - | 2 | 2 |

**Task-specific settings (Meta-World).** For **T–SAC**, we initialize the policy's log standard deviation as $\log \sigma = -5$. The replay buffer stores 5,000 segments of length 125 (i.e., $5,000 \times 125 = 625,000$ transitions). The sampler retrieves 4 segments of length 125 (i.e., $4 \times 125 = 500$ transitions).

**Table 6:** Hyperparameters for the Box Pushing Dense, Episode Length $T = 100$

| | PPO | gSDE | GTrXL | SAC | CrossQ | TCP | BBRL | TOP-ERL | T-SAC |
|---|---|---|---|---|---|---|---|---|---|
| number samples | 48000 | 80000 | 8000 | 8 | 1 | 152 | 152 | 4 | 4 * 100 |
| GAE $\lambda$ | 0.95 | 0.95 | 0.95 | n.a. | n.a. | 0.95 | n.a. | n.a. | n.a. |
| discount factor | 1.0 | 1.0 | 0.99 | 0.99 | 0.99 | 1.0 | 1.0 | 1.0 | 0.99 |
| $\epsilon_\mu$ | n.a. | n.a. | n.a. | n.a. | n.a. | 0.05 | 0.1 | 0.005 | n.a. |
| $\epsilon_\Sigma$ | n.a. | n.a. | n.a. | n.a. | n.a. | 0.0005 | 0.00025 | 0.0005 | n.a. |
| trust region loss coef. | n.a. | n.a. | n.a. | n.a. | n.a. | 1 | 10 | 1.0 | n.a. |
| optimizer | adam | adam | adam | adam | adam | adam | adam | adam | adamw |
| epochs | 10 | 10 | 5 | 1 | 1 | 50 | 20 | 15 | 20 |
| learning rate | 5e-5 | 1e-4 | 2e-4 | 3e-4 | 3e-4 | 3e-4 | 3e-4 | 3e-4 | 2.5e-4 |
| use critic | True | True | True | True | True | True | True | True | True |
| epochs critic | 10 | 10 | 5 | 1 | 1 | 50 | 10 | 30 | 100 |
| learning rate critic | 1e-4 | 1e-4 | 2e-4 | 3e-4 | 3e-4 | 1e-3 | 3e-4 | 5e-5 | 2.5e-5 |
| number minibatches | 40 | n.a. | n.a. | n.a. | n.a. | n.a. | n.a. | n.a. | n.a. |
| batch size | n.a. | 2000 | 1000 | 512 | 256 | n.a. | n.a. | 512 | 256 |
| buffer size | n.a. | n.a. | n.a. | 2e6 | 1e6 | n.a. | n.a. | 7000 | 20000 * 100 |
| learning starts | 0 | 0 | 0 | 1e5 | 5000 | 0 | 0 | 8000 | 5000 |
| temperature warmup | 0 | 0 | 0 | 0 | 0 | 0 | 0 | 0 | 0 |
| polyak_weight | n.a. | n.a. | n.a. | 5e-3 | 1.0 | n.a. | n.a. | 5e-3 | 5e-3 |
| SDE sampling frequency | n.a. | 4 | n.a. | n.a. | n.a. | n.a. | n.a. | n.a. | n.a. |
| entropy coefficient | 0 | 0.01 | 0 | auto | auto | 0 | 0 | 0 | 0 |
| normalized observations | True | True | False | False | False | True | False | False | False |
| normalized rewards | True | True | 0.1 | False | False | False | False | False | False |
| observation clip | 10.0 | n.a. | n.a. | n.a. | n.a. | n.a. | n.a. | n.a. | n.a. |
| reward clip | 10.0 | 10.0 | 10. | n.a. | n.a. | n.a. | n.a. | n.a. | n.a. |
| critic clip | 0.2 | 0.2 | 10. | n.a. | n.a. | n.a. | n.a. | n.a. | n.a. |
| importance ratio clip | 0.2 | 0.2 | 0.1 | n.a. | n.a. | n.a. | n.a. | n.a. | n.a. |
| hidden layers | [512, 512] | [256, 256] | n.a. | [256, 256] | [256, 256] | [128, 128] | [128, 128] | [256, 256] | [4 layers × 512] |
| hidden layers critic | [512, 512] | [256, 256] | n.a. | [256, 256] | [256, 256] | [256, 256] | [256, 256] | n.a. | n.a. |
| hidden activation | tanh | tanh | relu | tanh | tanh | leaky_relu | leaky_relu | leaky_relu | relu |
| orthogonal initialization | Yes | No | xavier | fanin | fanin | Yes | Yes | Yes | fanin |
| initial std | 1.0 | 0.05 | 1.0 | 1.0 | 1.0 | 1.0 | 1.0 | 1.0 | 1.0 |
| number of heads | - | - | 4 | - | - | - | - | 8 | 4 |
| dims per head | - | - | 16 | - | - | - | - | 16 | 64 |
| number of attention layers | - | - | 4 | - | - | - | - | 2 | 2 |
| MP type | n.a. | n.a. | value | n.a. | n.a. | ProDMP | ProDMP | ProDMP | n.a. |
| number basis functions | n.a. | n.a. | value | n.a. | n.a. | 8 | 8 | 8 | n.a. |
| weight scale | n.a. | n.a. | value | n.a. | n.a. | 0.3 | 0.3 | 0.3 | n.a. |
| goal scale | n.a. | n.a. | value | n.a. | n.a. | 0.3 | 0.3 | 0.3 | n.a. |

**Table 7:** Hyperparameters for the Box Pushing Sparse, Episode Length $T = 100$

| | PPO | gSDE | GTrXL | SAC | CrossQ | TCP | BBRL | TOP-ERL | T-SAC |
|---|---|---|---|---|---|---|---|---|---|
| number samples | 48000 | 80000 | 8000 | 8 | 1 | 76 | 76 | 4 | 4 * 100 |
| GAE $\lambda$ | 0.95 | 0.95 | 0.95 | n.a. | n.a. | 0.95 | n.a. | n.a. | n.a. |
| discount factor | 1.0 | 1.0 | 1.0 | 0.99 | 0.99 | 1.0 | 1.0 | 1.0 | 1.0 |
| | | | | | | | | | |
| $\epsilon_\mu$ | n.a. | n.a. | n.a. | n.a. | n.a. | 0.05 | 0.1 | 0.005 | n.a. |
| $\epsilon_\Sigma$ | n.a. | n.a. | n.a. | n.a. | n.a. | 0.0005 | 0.00025 | 0.0005 | n.a. |
| trust region loss coef. | n.a. | n.a. | n.a. | n.a. | n.a. | 1 | 10 | 1.0 | n.a. |
| | | | | | | | | | |
| optimizer | adam | adam | adam | adam | adam | adam | adam | adam | adamw |
| epochs | 10 | 10 | 5 | 1 | 1 | 50 | 20 | 15 | 20 |
| learning rate | 5e-4 | 1e-4 | 2e-4 | 3e-4 | 3e-4 | 3e-4 | 3e-4 | 3e-4 | 2.5e-4 |
| use critic | True | True | True | True | True | True | True | True | True |
| epochs critic | 10 | 10 | 5 | 1 | 1 | 50 | 10 | 30 | 100 |
| learning rate critic | 1e-4 | 1e-4 | 2e-4 | 3e-4 | 3e-4 | 3e-4 | 3e-4 | 5e-5 | 3.0e-4 |
| number minibatches | 40 | n.a. | n.a. | n.a. | n.a. | n.a. | n.a. | n.a. | n.a. |
| batch size | n.a. | 2000 | 1000 | 512 | 512 | n.a. | n.a. | 512 | 256 |
| buffer size | n.a. | n.a. | n.a. | 2e6 | 2e6 | n.a. | n.a. | 7000 | 20000 * 100 |
| learning starts | 0 | 0 | 0 | 1e5 | 1e5 | 0 | 0 | 400 | 2000 |
| temperature warmup | 0 | 0 | 0 | 0 | 0 | 0 | 0 | 0 | 0 |
| polyak_weight | n.a. | n.a. | 0 | 5e-3 | 1.0 | n.a. | n.a. | 5e-3 | 5e-3 |
| SDE sampling frequency | n.a. | 4 | n.a. | n.a. | n.a. | n.a. | n.a. | n.a. | n.a. |
| entropy coefficient | 0 | 0.01 | 0 | auto | auto | 0 | 0 | 0 | 0 |
| | | | | | | | | | |
| normalized observations | True | True | False | False | False | True | False | False | False |
| normalized rewards | True | True | 0.1 | False | False | False | False | False | False |
| observation clip | 10.0 | n.a. | False | n.a. | n.a. | n.a. | n.a. | n.a. | n.a. |
| reward clip | 10.0 | 10.0 | 10.0 | n.a. | n.a. | n.a. | n.a. | n.a. | n.a. |
| critic clip | 0.2 | 0.2 | 10.0 | n.a. | n.a. | n.a. | n.a. | n.a. | n.a. |
| importance ratio clip | 0.2 | 0.2 | 0.1 | n.a. | n.a. | n.a. | n.a. | n.a. | n.a. |
| | | | | | | | | | |
| hidden layers | [512, 512] | [256, 256] | n.a. | [256, 256] | [256, 256] | [128, 128] | [128, 128] | [256, 256] | [4 layers × 512] |
| hidden layers critic | [512, 512] | [256, 256] | n.a. | [256, 256] | [2048, 2048] | [256, 256] | [256, 256] | n.a. | n.a. |
| hidden activation | tanh | tanh | relu | tanh | relu | leaky_relu | leaky_relu | leaky_relu | leaky_relu |
| orthogonal initialization | Yes | No | xavier | fanin | fanin | Yes | Yes | Yes | fanin |
| initial std | 1.0 | 0.05 | 1.0 | 1.0 | 1.0 | 1.0 | 1.0 | 1.0 | 1.0 |
| number of heads | - | - | 4 | - | - | - | - | 8 | 4 |
| dims per head | - | - | 16 | - | - | - | - | 16 | 64 |
| number of attention layers | - | - | 4 | - | - | - | - | 2 | 2 |
| | | | | | | | | | |
| MP type | n.a. | n.a. | value | n.a. | n.a. | ProDMP | ProDMP | ProDMP | n.a. |
| number basis functions | n.a. | n.a. | value | n.a. | n.a. | 8 | 8 | 8 | n.a. |
| weight scale | n.a. | n.a. | value | n.a. | n.a. | 0.3 | 0.3 | 0.3 | n.a. |
| goal scale | n.a. | n.a. | value | n.a. | n.a. | 0.3 | 0.3 | 0.3 | n.a. |

**Table 8:** Hyperparameters for the Gymnasium MuJoCo, Episode Length $T = 1000$

|  | TD3 | CrossQ | SAC | T-SAC (Soft Copy) | T-SAC (Hard Copy) |
|---|---|---|---|---|---|
| number samples | 1 | 1 | 1 | 4 * 20 | 4 * 20 |
| GAE $\lambda$ | n.a. | n.a. | n.a. | n.a. | n.a. |
| discount factor | 0.99 | 0.99 | 0.99 | 0.99 | 0.99 |
| optimizer | adam | adam | adam | adamw | adamw |
| epochs | 1 | 1 | 1 | 12 | 12 |
| learning rate | 3e-4 | 1e-3 | 3e-4 | 3e-4 | 3e-4 |
| use critic | True | True | True | True | True |
| epochs critic | 1 | 3 | 1 | 60 | 60 |
| learning rate critic | 3e-4 | 1e-3 | 3e-4 | 3e-4 | 3e-4 |
| batch size | 256 | 256 | 256 | 256 | 256 |
| buffer size | 1e6 | 1e6 | 1e6 | 1e5 * 20 | 1e5 * 20 |
| learning starts | 5000 | 5000 | 5000 | 10000 | 10000 |
| temperature warmup | 0 | 0 | 0 | 10000 | 10000 |
| polyak_weight | 5e-3 | 1.0 | 5e-3 | 5e-3 | 1.0 |
| entropy coefficient | auto | auto | auto | auto | auto |
| hidden layers | [256, 256] | [256, 256] | [256, 256] | [256, 256] | [256, 256] |
| hidden layers critic | [256, 256] | [2048, 2048] | [256, 256] | n.a. | n.a. |
| hidden activation | relu | relu | relu | relu | relu |
| orthogonal initialization | fanin | fanin | fanin | fanin | fanin |
| initial std | 1.0 | 1.0 | 1.0 | 1.0 | 1.0 |
| number of heads | - | - | - | 4 | 4 |
| dims per head | - | - | - | 64 | 64 |
| number of attention layers | - | - | - | 2 | 2 |

**Task-specific settings (Gymnasium MuJoCo).** For **T–SAC**, we use the same task-agnostic settings across all Gymnasium MuJoCo tasks: the initial policy log-standard deviation is set to $-5$, and the target entropy is set to the SAC default $H_{\text{target}} = -\dim(\mathcal{A})$.

