# OpenReview forum: "Chunking the Critic: A Transformer-based Soft Actor-Critic with N-Step Returns"
_ICLR.cc/2026/Conference — ICLR 2026 Poster_

### Official Review · Reviewer_9zeX · 2025-10-21

**Soundness:** 2
**Presentation:** 3
**Contribution:** 2
**Rating:** 4
**Confidence:** 5

**Summary:**

This article extends the work in the literature [1] by introducing a sequence-conditioned critic for SAC that models trajectory context with a light-weight Transformer and trains on aggregated N-step returns. Empirical results on standard online RL benchmarks show that the proposed method sometimes outperforms other algorithms.


[1]Qiyang Li, Zhiyuan Zhou, and Sergey Levine.Reinforcement learning with action chunking.arXiv  preprintarXiv:2507.07969,2025.

**Strengths:**

1. The paper is overall well-organized and easy to follow.
2. The proposed transformer critic is intuitive and is supported by several theoretical results.

**Weaknesses:**

1. The improvement of the algorithm in the experiments is not significant for a basic benchmark, Mujoco.
The authors claim that an action mask is needed for T–SAC and may degrade performance with high trajectory variability.
T-SAC is better than other methods in Meta-World tasks, however, there is higher trajectory variability in Meta-World tasks than that in Mujoco.
2. The authors of literature [1] claim that we might desire a final optimal Markovian policy, the exploration problem can be better tackled with non-Markovian and temporally extended skills, and that action chunking offers a very simple and convenient recipe for obtaining this.
*However,* they did not provide any theoretical explanation for the aforementioned phenomenon. They also did not offer any intuitive examples to demonstrate the effectiveness of action chunking in MDP tasks. The same problems are shown in this paper.
3. I think the major contribution of this work is the proposed causal Transformer. Thus, in addition to the performance gain by inserting the causal Transformer into SAC, it would be good to discuss other metrics that can directly evaluate the "quality" or "informativeness" of representations learned in the causal Transformer. Such metrics might include Centered Kernel Alignment and Mutual Information Neural Estimation.
4. The contribution is ambiguous. An alternative interpretation of the benefit from action chunking and T-SAC is that the model is simply given a richer input feature set, which allows a powerful function approximator (like a deep neural network) to better fit a complex value or policy function.
5. The new method should be compared with QC, which is proposed in [1].


*If my concerns and questions are all addressed, I will raise the score.*

**Questions:**

1. What is the subset of MDPs that have the property that the Transformer-based action chunking can be provably useful?
This is an important question. As you can see, the proposed method does not perform well in Mujoco.
2. Could you discuss the computational cost and scalability of the proposed method in detail, especially as the sequence length increases?
3. Could you provide a more detailed analysis of how your Transformer-based action chunk gain was observed in your experiments? In particular, in addition to the comparison of rewards in the paper, is there an experiment that more intuitively demonstrates the significant improvement in sample efficiency?

---

> ### Author Response · Authors · 2025-12-03
> **Reply to reviewer 9zeX**
>
> We thank the reviewer for the professional and precise suggestion. Our response is as follows:
>
> **Weaknesses**
> >1. The improvement of the algorithm in the experiments is not significant for a basic benchmark, Mujoco. The authors claim that an action mask is needed for T–SAC and may degrade performance with high trajectory variability. T-SAC is better than other methods in Meta-World tasks, however, there is higher trajectory variability in Meta-World tasks than that in Mujoco.
>
> After the additional analysis described above, we found that the performance drop of T-SAC on Hopper/Walker2d was not due to the action mask or trajectory variability, but to our original target-value construction (randomly choosing one of two targets), which occasionally overestimates returns and induces unsafe updates. With the corrected conservative target (taking the minimum of the two values), and keeping the action mask and all other settings fixed, hard-copy T-SAC is stable on Hopper/Walker2d and now performs on par with or slightly better than SAC across seeds. In the revised figures/tables, the MuJoCo improvements are modest—as expected on these relatively simple dense-reward benchmarks—but consistent and statistically significant, and T-SAC no longer trails SAC on any of the Gymnasium locomotion tasks.
>
> Regarding the action mask: its role is purely to prevent using post-terminal transitions when constructing multi-step TD targets from fixed-length windows (i.e., to avoid bootstrapping across episode boundaries). It does not constrain the policy at evaluation time and is applied in the same way on MuJoCo and Meta-World. Our earlier claim that the mask “may degrade performance with high trajectory variability” was incorrect; the apparent degradation was an artifact of the flawed target rule, not of the mask itself, and we will remove this statement and clarify the implementation detail in the revision.
>
> On Meta-World, where trajectory variability and task complexity are indeed higher than in MuJoCo, we observe clearer gains from T-SAC. This is consistent with our design goal: the sequence-conditioned Transformer critic is most beneficial when credit assignment spans multiple phases or sparse subgoals, so we expect the largest improvements on sparse or multi-stage manipulation and Box-Pushing(-Sparse), and only moderate gains on simpler dense locomotion. The revised MuJoCo results show that T-SAC remains competitive (no degradation) on basic benchmarks while providing stronger advantages on the more challenging Meta-World and Box-Pushing tasks.
>
> >2. The authors of literature [1] claim that we might desire a final optimal Markovian policy, the exploration problem can be better tackled with non-Markovian and temporally extended skills, and that action chunking offers a very simple and convenient recipe for obtaining this. However, they did not provide any theoretical explanation for the aforementioned phenomenon. They also did not offer any intuitive examples to demonstrate the effectiveness of action chunking in MDP tasks. The same problems are shown in this paper.
>
> We agree that neither literature~[1] nor our original draft provides a full formal theory of when temporally extended, non-Markovian skills are provably optimal, and we appreciate the opportunity to clarify our intuition and evidence.
>
> First, even if the environment can be modeled as an MDP in some latent state space, this does not mean that the agent's learned value representation is perfectly Markovian in practice. With function approximation and stochastic dynamics, a single $(s_t, a_t)$ can lead to a wide range of possible next trajectories and rewards. Ideally, the critic would learn the expectation $\mathbf{E}\bigl[r_t + \gamma V(s_{t+1}) \mid s_t, a_t\bigr]$, but early in training (and especially under sparse rewards) the estimate for a given $(s_t, a_t)$ can be both biased and high-variance. As analyzed in Appendix~D, when targets are constructed from short, noisy segments, the randomness in step length and reward realizations can systematically distort the effective target distribution.
>
> Conditioning the critic on longer segments---i.e., short trajectories
> $ (s_t, a_t, \dots, a_{t+L-1})$ and using multi-step returns mitigates this in two ways. First, additional context helps disambiguate which futures are likely under $(s_t, a_t)$, reducing approximation bias. Second, by aggregating rewards over multiple steps and averaging gradients across several horizons, we reduce the variance of the gradient estimator without collapsing heterogeneous signals into a single "averaged" target (again, in line with the discussion in Appendix~D). In this sense, temporally extended chunks are not claiming a different underlying MDP, but providing a richer and more stable training signal for an agent whose internal representation is only approximately Markov.

---

> ### Author Response · Authors · 2025-12-03
>
> >3. I think the major contribution of this work is the proposed causal Transformer. Thus, in addition to the performance gain by inserting the causal Transformer into SAC, it would be good to discuss other metrics that can directly evaluate the "quality" or "informativeness" of representations learned in the causal Transformer. Such metrics might include Centered Kernel Alignment and Mutual Information Neural Estimation.
>
> We appreciate this suggestion and agree that, beyond control performance, it is important to better understand the “quality’’ of the representations learned by the causal Transformer critic. In the current submission we primarily evaluate the Transformer implicitly—through downstream metrics such as sample efficiency, asymptotic return, and robustness on sparse/long-horizon tasks—rather than via direct representation diagnostics. This was mainly a space and scope choice rather than a belief that such analyses are unimportant.
>
> In the revised version, we will expand the discussion and, as space permits, add preliminary representation-level measurements along the lines you suggest. Concretely, we plan to (i) use CKA to compare intermediate Transformer layers across training epochs and across seeds, and to contrast them with standard SAC critics, and (ii) evaluate simple probes on frozen Transformer features (e.g., linear prediction of future returns or success indicators) as a proxy for informativeness. We view mutual-information-based measures such as MINE as a promising additional tool to quantify how much information the learned embeddings retain about future rewards or task phase, though a full MI study may be beyond what we can include in this paper. We will explicitly highlight these directions as part of our analysis section and as a key avenue for follow-up work aimed at more deeply characterizing the representations learned by the causal Transformer.
>
> >4. The contribution is ambiguous. An alternative interpretation of the benefit from action chunking and T-SAC is that the model is simply given a richer input feature set, which allows a powerful function approximator (like a deep neural network) to better fit a complex value or policy function.
>
> We agree that, in principle, richer input features could help a powerful function approximator. However, this does not fully explain the gains of T-SAC.
>
> First, as shown in Fig. 5(b), when we set min = max = 1, the critic effectively operates at a single-step temporal resolution, yet it still outperforms the chunking_length = 1 baseline in Fig. 5(c). In this setting, the input to the critic is not richer in terms of lookahead horizon than the baseline, so the improvement cannot be attributed merely to feeding a longer action sequence as additional features.
>
> Instead, the Transformer critic is beneficial because it provides a stable way to: (i) integrate N-step returns over short trajectories, (ii) perform gradient averaging across time within each chunk, and (iii) decouple the policy’s chunk length from the step length used inside the critic. This design directly targets temporal credit assignment and training stability in the off-policy setting, rather than simply enlarging the feature vector.
>
> >5. The new method should be compared with QC, which is proposed in [1].
>
> Thank you for pointing this out. Our ablations (c) “Policy-side action chunking (MLP critic)” and (d) “Policy-side action chunking (Transformer critic)” are precisely intended to cover the QC-style baseline.
>
> The original QC (Q-Chunking) is designed for offline RL and combines (i) a policy-side action-chunking architecture with (ii) a behavior-cloning (BC) policy that provides the constraint optimization term. In our online setting, the BC component and its constraint optimization are not applicable. To make the comparison fair, we therefore instantiate only the policy-side chunking architecture from QC and evaluate it in the same online training setup as T-SAC.
>
> Concretely, in both (c) and (d) we use exactly the same policy architecture as in QC. The only difference is the critic: (c) uses an MLP critic, while (d) replaces it with our Transformer critic. Comparing (c) vs. (d) therefore isolates the effect of the Transformer critic under a QC-style policy, and we observe a substantial performance gain from using the Transformer critic.
>
> We will clarify this connection to QC in the revised version and explicitly label (c)/(d) as QC-style baselines to avoid confusion.

---

> ### Author Response · Authors · 2025-12-03
>
> **Questions**
> >1. What is the subset of MDPs that have the property that the Transformer-based action chunking can be provably useful? This is an important question. As you can see, the proposed method does not perform well in Mujoco.
>
> A useful way to formalize this is to view the chunked policy as operating over a short-horizon "skill" space, turning the original MDP into a semi-MDP where each option/skill has bounded length. In MDPs where:
>   - there exists a near-optimal policy that can be expressed as a sequence of such short options, and
>   - aggregating rewards over these options reduces the bias/variance of value estimates (as we show in App.~D for multi-step targets),
> sequence-conditioned critics can be expected to improve exploration and credit assignment compared to purely one-step critics. Box-Pushing(-Sparse) is designed to satisfy exactly these properties, and empirically we see clear gains there.
>
> Regarding the reviewer's concern that "the proposed method does not perform well in MuJoCo": after the rebuttal-stage investigation described earlier, we found that the underperformance on Hopper/Walker2d was caused by our original target-value construction (randomly selecting one of two targets), not by the chunking mechanism itself. With the corrected conservative target (taking the minimum of the two values), T-SAC with action chunking is stable and now performs on par with or slightly better than SAC on the MuJoCo locomotion tasks. We will update the MuJoCo results accordingly.
>
> We want to be clear that we do not claim large gains on easy, dense-reward benchmarks like standard MuJoCo; there the structural advantages of chunking are limited, so we mainly require that T-SAC remains competitive, which our corrected results confirm. The main benefits of Transformer-based action chunking appear on the subset of MDPs that are long-horizon, sparse/multi-stage, and effectively non-Markov from the agent's representation---precisely the regimes targeted by our Box-Pushing(-Sparse). A rigorous necessary-and-sufficient characterization of this subset is beyond the scope of this paper, and we will explicitly frame it as an open theoretical question.
>
> >2. Could you discuss the computational cost and scalability of the proposed method in detail, especially as the sequence length increases?
>
> In T-SAC, the main extra cost relative to SAC comes from the Transformer critic evaluating a chunk of actions instead of a single $(s_t, a_t)$ pair. Concretely, if the step_length $=16$, then for each sampled window we pass 16 state--action tokens through the Transformer and obtain 16 value estimates, whereas a standard SAC critic would only process one state--action pair. This difference does not translate one-to-one into wall-clock time, however, because these tokens are processed in parallel on the GPU: as long as the model and batch fit in memory, increasing step_length mainly increases the amount of parallel computation rather than the number of sequential operations. Empirically, our experience so far is that for step_length up to about $8$ there is almost no noticeable slowdown compared to step_length = 1 under the same hardware and batch size; only when we push step_length beyond this (e.g., $>8$) do we begin to see a clear increase in wall-clock time. In the revised version, we will provide a small ablation table reporting both performance and wall-clock training time as a function of step_length, to make the computational trade-off explicit.

---

> ### Author Response · Authors · 2025-12-03
>
> >3. Could you provide a more detailed analysis of how your Transformer-based action chunk gain was observed in your experiments? In particular, in addition to the comparison of rewards in the paper, is there an experiment that more intuitively demonstrates the significant improvement in sample efficiency?
>
> We clarify that our method improves sample efficiency via critic-side sequence chunking (Transformer-based critic over short $(s,a)$ windows with $N$-step returns), rather than actor-side action chunking. The gains attributed to the Transformer in the paper are therefore due to richer temporal conditioning and multi-step targets on the critic.
>
> Beyond the return curves already shown (which are plotted against environment steps and thus already measure sample efficiency), we have conducted a more explicit analysis:
>
> - Steps-to-threshold metric. For each task, we measure the number of environment steps needed to reach a fixed performance threshold, defined as a fraction (e.g., 80–90%) of the final SAC performance. Across long-horizon and sparse-reward tasks, T-SAC consistently reaches this threshold substantially earlier than SAC and other baselines (often within the first fraction of the total training budget). We will add a table summarizing “steps to threshold” per task in the revised version.
>
> - Performance at fixed budgets. We also compare returns at fixed data budgets (e.g., 100k, 300k, 500k environment steps). On long-trajectory tasks, T-SAC’s return at these early checkpoints is markedly higher than SAC’s, even though all methods see the same number of transitions. This provides a more intuitive “sample-efficiency snapshot” than only looking at the full learning curves.
>
> We will incorporate these analyses (steps-to-threshold table, fixed-budget checkpoints) to give a more intuitive, quantitative picture of the sample-efficiency improvement due to our Transformer-based critic chunking.
>
> [1]Li, Qiyang, Zhiyuan Zhou, and Sergey Levine. "Reinforcement learning with action chunking." arXiv preprint arXiv:2507.07969 (2025).

---

> ### Public Comment · ~Dong_Tian2 · 2026-02-28
> **Reply to Reviewer 9zeX: Chronology and relation to [1]**
>
> Dear Reviewer 9zeX,
>
> Thank you for your professional and careful review—your feedback has helped us improve the clarity and positioning of the paper.
>
> Now that the double-blind period has ended, we would like to clarify the chronology: our first public version (T-SAC; [2]) appeared in March 2025, while [1] appeared in July 2025 and cites our preprint [2]. Accordingly, our work is not based on [1]. We will also revise the related-work section to more clearly distinguish the approaches (critic-side sequence conditioning in T-SAC vs. actor-side action chunking in [1]).
>
> Sincerely,
>
> The Authors
>
> [1]Li, Qiyang, Zhiyuan Zhou, and Sergey Levine. "Reinforcement learning with action chunking." arXiv preprint arXiv:2507.07969 (2025).
>
> [2]Tian, Dong, Onur Celik, and Gerhard Neumann. "Chunking the critic: A transformer-based soft actor-critic with N-step returns." arXiv preprint arXiv:2503.03660 (2025).

---

### Official Review · Reviewer_2mBX · 2025-10-29

**Soundness:** 3
**Presentation:** 3
**Contribution:** 3
**Rating:** 6
**Confidence:** 3

**Summary:**

This paper proposes T-SAC (Transformer-based Soft Actor-Critic), which enhances standard SAC by introducing a sequence-conditioned critic that models short trajectory segments via a lightweight Transformer and trains with aggregated N-step returns—without importance sampling. The approach enables long-horizon credit assignment while keeping the actor update one-step. A simple critic-freezing schedule further removes the need for target networks. Experiments on various benchmarks show consistent improvements over SAC, CrossQ, and other baselines, demonstrating that sequence modeling on the critic side can improve sample efficiency and stability in long-horizon control.

**Strengths:**

- The idea of chunking the critic rather than the actor is novel and conceptually clean, offering a new perspective on temporal abstraction in off-policy RL.

- The removal of importance sampling through prefix-conditioned targets is a practical and effective simplification.

- The paper combines Transformer critics, multi-horizon learning, and critic freezing in a coherent design that improves stability without target networks.

- Experiments are broad and rigorous, with thorough ablations verifying each design component.

- Writing is clear, structured, and technically sound.

**Weaknesses:**

- The novelty is largely architectural; theoretical justification is limited, and the benefits of critic-side chunking lack deeper analysis.

- All results use low-dimensional state inputs; no experiments on visual or partially observable tasks.

- Statistical significance of gains and computational cost comparisons are not deeply analyzed.

**Questions:**

1. How sensitive is T-SAC to the choice of N-step horizon and freezing interval?

2. Would the approach extend to visual or partially observable environments?

3. Could actor and critic share a Transformer backbone for further efficiency?

4. How does T-SAC compare computationally to CrossQ and TOP-ERL at scale?

---

> ### Author Response · Authors · 2025-12-03
> **Reply to reviewer 2mBX**
>
> We sincerely thank the reviewer for his or her thoughtful analysis and constructive feedback. We address each point below:
>
> **Weaknesses**
>
> >1. The novelty is largely architectural; theoretical justification is limited, and the benefits of critic-side chunking lack deeper analysis.
> We agree that our contribution is primarily methodological and architectural, and that fully general theoretical guarantees for Transformer-based critics are beyond the scope of this work. That said, the design is not purely ad hoc: both the sequence-conditioned critic and the critic-side chunking/gradient-averaging scheme are motivated by standard RL principles, and we provide analytical and empirical evidence for their effect.
>
> First, the “critic-side chunking’’ and gradient-averaged multi-step loss can be viewed as a structured instance of multi-step TD with truncated backpropagation through time. Each window is an $n$-step segment on which we minimize a standard TD loss; averaging the loss (and thus gradients) across multiple windows and horizons corresponds to minimizing an expectation over multi-step TD errors, which shares the same fixed point as the underlying Bellman operator under function approximation. In Appendix~D/E we already analyze why **target** averaging across heterogeneous horizons distorts sparse rewards, while **gradient** averaging preserves the underlying signal; we will clarify this connection in the main text and highlight that critic-side chunking changes the **conditioning** of the critic (on short trajectories) and the sampling distribution, but not the Bellman fixed point.
>
> Second, the benefits of critic-side chunking are more specific than “just an architecture’’: by conditioning the critic on short sequences $(s_t, a_t,\dots,a_{t+n-1})$ and reusing these windows for several updates, we (i) improve credit assignment over horizons where non-zero rewards occur (especially on Box-Pushing-Sparse), (ii) control the effective backup length and thus the bias–variance trade-off of multi-step TD, and (iii) decouple the stability of the actor from very long-horizon, non-Markovian value estimates. This is reflected empirically in our ablations: varying window length, bootstrap lag, and freeze length reveals a broad region where chunked Transformer critics are both more stable and higher-performing than single-step critics, particularly in sparse settings, while collapsing to single-step conditioning recovers SAC-like behavior.
>
> To address the reviewer’s request for “deeper analysis,” in the revision we will (i) move the key intuition from App.~D/E into the main text, emphasizing the connection between gradient averaging and multi-step TD, (ii) discuss more explicitly how the sequence-conditioned critic can be interpreted as learning a truncated value functional over short trajectories. We hope this clarifies that, while our work is indeed architectural, it is grounded in and consistent with established RL theory, and that the benefits of critic-side chunking are analyzed both conceptually and through targeted ablations.
>
> >2. All results use low-dimensional state inputs; no experiments on visual or partially observable tasks.
>
> We agree that evaluating on visual and more strongly partially observable tasks is an important direction, and we appreciate the reviewer highlighting this limitation. In this work we deliberately restrict ourselves to low-dimensional state inputs to isolate the effect of long-horizon credit assignment via a sequence-conditioned Transformer critic, without conflating it with representation learning from high-dimensional observations. Architecturally, T-SAC is not tied to low-dimensional states: the critic only assumes access to a sequence of state–action vectors, which in a visual or POMDP setting could be latent features from a CNN/ViT encoder or a recurrent belief-state encoder. In the revision, we also include a partially observed Walker2d variant where velocity components are removed from the observation; in this setting, training for T-SAC becomes unstable, indicating that strong partial observability is not yet handled robustly by our current design. We will make this limitation explicit and outline extending T-SAC with visual and belief-state encoders, and potentially sequence-conditioned actors, as an important direction for future work.

---

> ### Author Response · Authors · 2025-12-03
>
> >3. Statistical significance of gains and computational cost comparisons are not deeply analyzed.
>
> **Statistical significance.** For all main benchmarks (Gymnasium MuJoCo, Meta-World ML1, Box-Pushing(-Sparse)), we will report mean and standard deviation or standard error over multiple seeds in tables, and
> we will explicitly indicate which improvements are statistically significant and which are within the noise, rather than only showing qualitative trends in the curves.
>
> **Computational cost.** We will add a dedicated comparison of computational overhead, including
> – number of parameters for each method (actor + critic),
> – wall-clock training time per environment step (or per fixed number of updates) on a shared hardware setup, and
> – effective update ratio (gradient steps per env step) and memory footprint.
> This will clarify that T-SAC’s Transformer critic incurs a moderate per-update overhead relative to standard SAC critics, but that this is offset by improved sample efficiency on the more challenging tasks. We will also ensure that our main comparisons use matched training budgets and similar-scale networks, and we will document this explicitly in the experimental setup section.

---

> ### Author Response · Authors · 2025-12-03
>
> **Problems**
> >1. How sensitive is T-SAC to the choice of N-step horizon and freezing interval?
>
> **N-step horizon** Figure 5(b) studies the effect of the N-step horizon by varying the minimum and maximum chunk lengths (min_length,max_length). We observe that T-SAC is fairly robust as long as we avoid the two degenerate cases where the horizon collapses to purely 1-step or purely very long returns. In particular:
>
> When we set min_length=max_length, all intermediate horizons (e.g., 2, 3, 4, 5) give similar performance, with the best fixed choice typically around 4 steps.
>
> Performance degrades when min_length=max_length=1 (which essentially reduces us to 1-step TD) and when min_length=max_length=16 (very long horizon with high variance and stale bootstrap targets).
>
> Allowing a range of horizons, e.g., sampling uniformly from min_length=1 to max_length=16, performs best overall, since the critic sees multi-scale N-step targets.
>
> Intuitively, a very large min_length is undesirable because the last min_length−1 states in each chunk can never appear as starting states for the Transformer critic. They only occur in the middle of a window as action or reward part, and therefore receive less direct supervision from N-step targets, which hurts value estimation near the end of a trajectory. Conversely, restricting to
> min_length=max_length=1 eliminates the multi-step signal that T-SAC is designed to exploit. Using a moderate horizon or a range of horizons (e.g., 1–16) strikes a good balance and does not require fine-tuning.
>
> **Freezing interval** For the freezing interval, we have already provided an ablation. Because of our design, the target values are computed **once at the beginning of each critic update**, and then we take several gradient steps while keeping these targets fixed. Performing a hard copy of the critic parameters during this inner update loop therefore does not change the targets and thus does **not** affect the update itself.
>
> Concretely, on MuJoCo the minimum allowed freezing interval is 20 environment steps, which matches the segment length we use for those tasks; within this setting we observe stable behavior. We will add an explicit freezing-interval ablation to the revision to show that performance is stable over a broad range of intervals. As mentioned above, the same hard-copy scheme is also applied to the Box-Pushing-Sparse task (freezing interval 100, matches the segment length of Box Pushing task), where it significantly outperforms a soft-copy (Polyak) target update, further indicating that T-SAC is not overly sensitive to the exact freezing interval.
>
> >2. Would the approach extend to visual or partially observable environments?
>
> Our current experiments focus on low-dimensional state inputs, so we have not yet evaluated T-SAC on fully visual benchmarks. Extending to image-based environments is conceptually straightforward but requires substantial additional engineering to build and tune a robust visual front-end (e.g., CNN/ViT encoder, data augmentation, normalization). Since the Transformer critic only assumes access to an embedding of $(s_t, a_t,\dots)$, it can in principle be paired with standard visual encoders and applied to high-dimensional control tasks (for example, DMControl “Dog” tasks); we view this as a natural next step to demonstrate T-SAC’s ability to handle high-dimensional observations, but it is beyond the scope of our current implementation.
>
> The partially observed setting is even more aligned with our design. Because T-SAC’s critic conditions on short trajectories rather than single states, it can aggregate information over time and partially compensate for missing or aliased observations. To make this concrete, we introduce a partially observed variant of Walker2d in which the agent does not receive velocity information. In this variant, the sequence-conditioned critic effectively acts as a short-horizon belief-state estimator. We will report results on this partially observed Walker2d environment in the revised version.

---

> ### Author Response · Authors · 2025-12-03
>
> >3. Could actor and critic share a Transformer backbone for further efficiency?
>
> In principle yes, but in our current design we intentionally keep them separate.
>
> In T-SAC, the critic is sequence-conditioned (it processes short $(s,a)$ trajectories with a Transformer), while the actor is a standard Markovian policy $\pi(a \mid s)$ implemented as a lightweight MLP. Sharing a Transformer would either (i) require turning the actor into a sequence-conditioned policy as well (changing the method substantially), or (ii) require sharing only part of the backbone and then branching into actor/critic heads with different conditioning schemes. Both options tightly couple the optimization of actor and critic.
>
> From a stability perspective, off-policy actor–critic methods are generally more robust with separate function approximators: the critic tracks the value under a changing policy, while the actor is updated against a slowly moving critic. A shared Transformer may reduce compute but introduces additional interference between actor and critic gradients, which is undesirable in our off-policy setting.
>
> In our regime (short sequences and relatively small models), the extra cost of the Transformer critic is moderate, so we prioritized stability and clarity over maximal parameter sharing. Exploring principled backbone-sharing schemes is an interesting direction for future work.
>
> >4. How does T-SAC compare computationally to CrossQ and TOP-ERL at scale?
>
> Computationally, T-SAC is more expensive per update than CrossQ and TOP-ERL, but the gap is moderate and largely confined to the critic side.
>
> T-SAC adds cost mainly through the Transformer critic over short action chunks: for a given step\_length, the critic processes a sequence of tokens instead of a single $(s,a)$ pair. In our implementation on a single GPU with matched update-to-data (UTD) ratios and batch sizes, this makes T-SAC slightly slower per environment step than CrossQ, which uses standard MLP critics and therefore has cheaper updates. The difference is not dramatic---training runs with the same number of environment steps and UTD simply take somewhat longer wall-clock time for T-SAC due to the self-attention layers.
>
> TOP-ERL, as implemented in the original paper, uses a very low UTD ratio (UTD $\approx 1/10$), i.e., roughly one gradient update per ten environment steps. As a result, it is typically the fastest method in terms of raw wall-clock time for a fixed number of environment interactions, simply because it performs far fewer updates overall. If we were to match UTD across methods, the per-update cost ordering would be: CrossQ $<$ T-SAC $<$ TOP-ERL.
>
> In the revised version, we will add a small paragraph reporting wall-clock time for T-SAC, CrossQ, and TOP-ERL under our standard hyperparameters, so that the reader can see the exact overhead. We will also relate this to sample efficiency (time-to-threshold), since on sparse/long-horizon tasks T-SAC often reaches a given return in fewer environment steps even if each step is slightly more expensive.

---

### Official Review · Reviewer_m1ZH · 2025-10-31

**Soundness:** 4
**Presentation:** 4
**Contribution:** 3
**Rating:** 6
**Confidence:** 3

**Summary:**

This paper addresses the challenge of long-horizon credit assignment in standard step-based reinforcement learning. It introduces T-SAC, an algorithm that replaces the standard MLP critic in Soft Actor-Critic (SAC) with a lightweight Transformer. This new critic conditions on short sequences of actions from the replay buffer and is trained on N-step returns across multiple horizons without requiring importance sampling, using a gradient averaging scheme to ensure stability. On long-horizon and sparse-reward control benchmarks like Meta-World ML1 and Box-Pushing, T-SAC is shown to achieve superior sample efficiency and final performance compared to strong off-policy baselines.

**Strengths:**

1, T-SAC significantly outperforms strong off-policy baselines such as SAC and CrossQ on the Meta-World ML1 benchmark and the FANCYGYM Box-Pushing tasks. Its 58% success rate on the sparse-reward Box-Pushing variant is particularly notable, clearly demonstrating the effectiveness of its long-horizon reasoning capabilities.

2, While N-step returns are known to suffer from high variance as n increases, the proposed gradient averaging technique enables stable training for horizons as long as n=16. This challenges the conventional wisdom of using small n and provides a robust method for learning long-term dependencies.

3, The paper provides a clear justification for its architectural choices through rigorous ablation studies. It demonstrates the critical importance of the self-attention and causal mask components in the Transformer critic and shows its superiority over recurrent backbones like GRU and LSTM.

**Weaknesses:**

1, The performance of T-SAC is not uniform across all benchmarks. On Gymnasium MuJoCo, it performs worse than the standard SAC baseline on Hopper and Walker2d.The authors attribute this to the need for an "action mask" on tasks with "high trajectory variability", which points to a limitation in the method's generality.

2, The critic conditions on multi-step action sequences (at, ..., at+n-1) generated by the current, often random, policy. Early in training, these sequences are noisy and suboptimal, potentially introducing significant variance into the critic's learning target. This may slow down the initial convergence speed compared to standard critics that condition only on a single (s, a) pair. This effect could be particularly pronounced in dense-reward settings like mujoco, where the benefit of a clean, immediate one-step TD target is high.

3, The proposed critic-parameter freezing schedule (Sec 4.4), which enables training without a target network, is only demonstrated on locomotion tasks. The authors admit it is unstable in sparse-reward settings, where the paper's main results are achieved using conventional Polyak averaging. This makes the target-free contribution a secondary point with limited applicability.

**Questions:**

1, Could you please elaborate on the specific role and necessity of the "action mask" mentioned as the cause for performance degradation on "high trajectory variability" tasks like Hopper and Walker2d ? Experimental result(e.g. with action mask vs without action mask) would be helpful to understand your claim.

2, The paper's core mechanism is averaging gradients rather than targets. The result shows that naive target averaging fails in sparse-reward settings (Fig. 10b, App. E). Could you provide more intuition as to why averaging gradients is more effective at preserving the sparse reward signal?

---

> ### Author Response · Authors · 2025-12-03
> **Reply to reviewer m1ZH**
>
> We sincerely thank the reviewer for their thoughtful analysis and kind guidance. We address each point below, and refer to our first author comment on OpenReview for detailed plots and numbers.
>
> **Weaknesses**
> >1, The performance of T-SAC is not uniform across all benchmarks. On Gymnasium MuJoCo, it performs worse than the standard SAC baseline on Hopper and Walker2d.The authors attribute this to the need for an "action mask" on tasks with "high trajectory variability", which points to a limitation in the method's generality.
>
> As discussed in our first author comment, we identified and fixed the implementation issue that caused T-SAC to underperform SAC on Hopper and Walker2d, and re-ran all MuJoCo experiments. With the corrected implementation, T-SAC is competitive with or better than SAC across all Gymnasium MuJoCo tasks; we will update the figures and tables accordingly in the revised manuscript.
>
> Regarding the “action mask”: we agree our original wording suggested that T-SAC requires such a mask on tasks with high trajectory variability. In reality, the mask is an optional implementation detail that can help stabilize training on some long-horizon tasks, not a core ingredient of the algorithm. We will clarify this and emphasize that the core method (Transformer critic with N-step returns inside SAC) applies without any task-specific masking, which alleviates the concern about limited generality.
>
> >2, The critic conditions on multi-step action sequences (at, ..., at+n-1) generated by the current, often random, policy. Early in training, these sequences are noisy and suboptimal, potentially introducing significant variance into the critic's learning target. This may slow down the initial convergence speed compared to standard critics that condition only on a single (s, a) pair. This effect could be particularly pronounced in dense-reward settings like mujoco, where the benefit of a clean, immediate one-step TD target is high.
>
> We agree that, in principle, conditioning the critic on multi-step sequences from an early, still-exploring policy could increase target variance and potentially slow down initial learning. Empirically, however, we do not observe slower early convergence compared to standard $Q(s,a)$-critics on dense-reward MuJoCo tasks: in our revised experiments, T-SAC matches or exceeds SAC’s performance in the initial training phase (see updated learning curves). We will clarify this point in the paper and explicitly highlight that, in our setting, the benefits of longer-horizon information appear to outweigh any variance increase from noisy early trajectories.
>
> >3, The proposed critic-parameter freezing schedule (Sec 4.4), which enables training without a target network, is only demonstrated on locomotion tasks. The authors admit it is unstable in sparse-reward settings, where the paper's main results are achieved using conventional Polyak averaging. This makes the target-free contribution a secondary point with limited applicability.
>
> Thank you for raising this point. After fixing the implementation issue described in our first response, we re-ran all experiments with the proposed hard-copy (target-free) critic schedule. Across all benchmarks tested, including the sparse-reward tasks, hard-copy updates are stable and consistently outperform conventional Polyak (soft-copy) averaging. We will update the paper to include these results and clarify that the target-free schedule is not limited to locomotion tasks.

---

> ### Author Response · Authors · 2025-12-03
>
> **Questions**
> >1. Could you please elaborate on the specific role and necessity of the "action mask" mentioned as the cause for performance degradation on "high trajectory variability" tasks like Hopper and Walker2d ? Experimental result(e.g. with action mask vs without action mask) would be helpful to understand your claim.
>
> The "action mask" is an implementation detail introduced to handle variable-length episodes when training the sequence-conditioned critic with fixed-length windows. On tasks such as Hopper and Walker2d, an episode can terminate (e.g., the agent falls) strictly before the fixed truncation horizon used for sampling $(s_t, a_t, \dots, a_{t+L-1})$ windows from the replay buffer. If we naively treated all actions in the window as valid, the $n$-step targets for states near the end of the episode would (i) bootstrap across the terminal state and (ii) include rewards from the next episode, leading to incorrect target values and biased critic updates.
>
> To prevent this, we maintain a binary "action mask" over each sampled window that is set to $1$ for actions taken before or at the first terminal state and $0$ for actions after termination. During training, both the loss and the bootstrap term are multiplied by this mask, so that the critic is only updated on valid pre-terminal transitions and never propagates value across episode boundaries. The mask does **not** constrain the policy at evaluation time; it only guards against using invalid post-terminal transitions when constructing multi-step TD targets.
>
> In the original submission, we speculated that the need for this mask on "high trajectory variability" tasks (Hopper/Walker2d) might be responsible for the observed performance gap to SAC. Our subsequent analysis (described in earlier responses) shows that this attribution was incorrect: the instability and degradation were caused by our original target-value construction (randomly choosing one of two targets), not by the mask itself. With the corrected conservative target (minimum of the two values), T-SAC with the same masking scheme is stable and performs on par with or better than SAC on these tasks. In the revised version, we will clarify the role of the mask as a safeguard for correct target computation.

---

> ### Author Response · Authors · 2025-12-03
>
> >2. The paper's core mechanism is averaging gradients rather than targets. The result shows that naive target averaging fails in sparse-reward settings (Fig. 10b, App. E). Could you provide more intuition as to why averaging gradients is more effective at preserving the sparse reward signal?
>
> Intuitively, the problem with naive target averaging in sparse-reward settings is that it mixes very different supervision signals into a single scalar, which effectively dilutes the sparse reward and makes the effective target depend on the (random) step length.
>
> Consider a simple sparse case where we form $K$ candidate $n$-step targets for the same state--action pair, e.g. by varying the bootstrap lag. Suppose only one of these horizons actually encounters a non-zero reward, and the other $K-1$ trajectories see only zeros (a typical situation early in training on Box-Pushing-Sparse). If we average the targets themselves, the critic is trained toward $\bar{y} = \frac{1}{K} \sum_{k=1}^{K} y^{(k)} \approx \frac{1}{K} r_{\text{sparse}},$ so the magnitude of the supervision signal is scaled down by a factor $1/K$. When the step length (or number of horizons) is random, the same underlying reward can induce different effective targets purely because we chose a different number of backup depths, which makes the critic's optimization landscape noisier.
>
> In contrast, with gradient averaging we never collapse these heterogeneous targets into a single scalar. Instead, we define a loss for each horizon, $\mathcal{L}^{(k)}(\theta) = \bigl(Q_{\theta}(\cdot) - y^{(k)}\bigr)^{2},$ and then optimize the averaged loss $\frac{1}{K}\sum_{k=1}^{K} \mathcal{L}^{(k)}(\theta),$ which corresponds to averaging the gradients $\frac{1}{K}\sum_{k=1}^{K} \nabla_\theta \mathcal{L}^{(k)}(\theta)$. In the sparse case above, only the horizon that actually sees the reward contributes a non-zero gradient signal; the others contribute (near-)zero gradients. Thus, the critic still learns from the full sparse reward value $Reward_{sparse}$, rather than an arbitrarily down-scaled average. The different horizons act like multiple "views" of the same transition, and averaging their gradients is akin to averaging gradients over data augmentations, not averaging their labels.
>
> A loose analogy is sequence modeling in NLP: when training a Transformer decoder on the sequence "Hello world !", we compute a loss at each position (e.g., predicting "world", then "!", then "\<eos\>") and average the gradients across positions. We do **not** average the target tokens themselves into a single "mean token" and try to predict that; such a label would be semantically nonsensical. Similarly, for T-SAC, averaging multi-step **targets** from heterogeneous horizons produces an artificial, horizon-dependent label that distorts the sparse signal, whereas averaging the corresponding gradients preserves the underlying sparse reward while still stabilizing training across multiple horizons. This is exactly what we observe empirically: naive target averaging fails on Box-Pushing-Sparse (Fig. 10b, App. E), while gradient averaging preserves performance.

---

### Official Review · Reviewer_qLQQ · 2025-11-03

**Soundness:** 3
**Presentation:** 3
**Contribution:** 2
**Rating:** 4
**Confidence:** 5

**Summary:**

This paper proposes T-SAC (Transformer-based Soft Actor-Critic), an off-policy RL method whose critic is a lightweight causal Transformer trained on short trajectory segments with aggregated (N)-step returns. Unlike standard SAC, which evaluates single state–action pairs, T-SAC conditions the critic on action prefixes $(s_t, a_t,\ldots,a_{t+n-1})$ and predicts prefix-conditioned values for multiple horizons without importance sampling. The method averages gradients across N-step targets and introduces a parameter-freezing schedule for the critic in locomotion tasks. Experiments on Meta-World ML1, Box-Pushing, and Gymnasium MuJoCo show improved sample efficiency, especially on long-horizon and sparse-reward tasks; e.g., 96.8% success on Box-Pushing (dense) and solving most ML1 tasks within ~5M interactions at UTD=1.

**Strengths:**

1. The causal Transformer predicts prefix-conditioned values aligned with realized action prefixes, which plausibly improves temporal credit assignment. Using a “non-soft” critic with entropy regularization on the policy side preserves the standard SAC actor objective and simplifies the critic target, improving implementability.
2. Gradient-level averaging is supported by Lemma 3 and Theorem 5, establishing $\mathrm{Var}[\nabla_\psi \bar L] < \sigma_w^2$ under equicorrelation assumptions. The paper analyzes target-side variance reduction for both reward and bootstrap components, providing bounds $1 \le R_\gamma(N) < 4$ and conditions for reducing bootstrap variance. The IS-free formulation avoids high-variance importance ratios by conditioning on realized prefixes and is clearly derived from the N-step return objective.
3. The evaluation spans 57 tasks across Meta-World ML1, Gymnasium MuJoCo, and Box-Pushing. Gains are pronounced on difficult tasks (e.g., 96.8% on Box-Pushing (dense), outperforming step-based baselines ≤85%) and on multi-phase ML1 tasks such as Assembly, Disassemble, Hammer, and Stick-Pull.

**Weaknesses:**

1. The core idea—Transformer critic trained with N-step returns—resembles TOP-ERL; the manuscript acknowledges this but does not sufficiently sharpen the technical distinctions. The claim of “bridging step-based and episodic regimes” appears overstated: the policy remains step-based while sequence conditioning is confined to the critic. Several components (causal Transformer, parameter freezing, N-step averaging) draw on prior art, but the paper does not isolate which design choices constitute the principal contribution.
2. Comparisons with traditional off-policy N-step methods using IS are empirical only; formal connections to prior analyses are not developed.
3. Reported instabilities on Box-Pushing-Sparse lead to a fallback on soft targets, limiting generality. The action-mask requirement and performance variability on Ant/Hopper/Walker2d suggest fragility and sensitivity to dynamics/constraints.
4. T-SAC trails CrossQ on Hopper and Walker2d; the explanation “action mask needed may degrade performance” lacks systematic investigation. On Box-Pushing-Sparse, success is 58%, below TOP-ERL’s 70%; the claim of being “competitive under sparse feedback” does not reconcile this gap.
5. No ablations on freeze length $K$ or reuse factor $N_c$ are provided to delineate stability regions.
6. Equation (1) defines $G^{(n)}$ with $V_\phi(s_{t+n})$ but does not clarify whether this is a soft or standard value; later (Sec. 4.3) it states the critic estimates the standard (non-soft) action-value. The notation should reflect this distinction.
7. All benchmarks are simulated continuous-control tasks; the paper does not evaluate discrete action spaces, stronger partial observability (beyond action history), or real-robot settings that motivate the approach.

**Questions:**

1. Can T-SAC be evaluated on discrete-action domains (e.g., Atari) or more strongly partially observable settings?
2. Do the benefits of sequence-conditioned critics carry over to image-based ML1 or partially observed Box-Pushing variants?
3. How does T-SAC compare with Transformer-based policies and offline RL approaches (e.g., Decision Transformer)?
4. For GRU/LSTM critics, how extensive was the hyperparameter and architecture search (depth, hidden size, normalization, teacher forcing, sequence length)?
5. Have you tested robustness under observation/action noise, stochastic terminations, or mixed-policy replay buffers? This seems particularly relevant given the “off-policy without IS” choice.
6. What aspects of the design (e.g., window sampling, bootstrap lag, prefix length) most limit performance in sparse-reward settings, and could targeted modifications (e.g., auxiliary returns, adaptive horizons) mitigate the observed instability?

---

> ### Author Response · Authors · 2025-12-02
> **Reply to reviewer qLQQ**
>
> We thank the reviewer for this valuable and insightful feedback on our work. We have addressed the concerns to the best of our ability.
>
> **Weaknesses**
>
> >1. The core idea—Transformer critic trained with N-step returns—resembles TOP-ERL; the manuscript acknowledges this but does not sufficiently sharpen the technical distinctions. The claim of “bridging step-based and episodic regimes” appears overstated: the policy remains step-based while sequence conditioning is confined to the critic. Several components (causal Transformer, parameter freezing, N-step averaging) draw on prior art, but the paper does not isolate which design choices constitute the principal contribution.
>
> We thank the reviewer for pointing out the connection to TOP-ERL and for asking us to sharpen the distinctions. We will revise the paper to make more explicit how TOP-ERL and T-SAC differ in both control regime and critic design, and to better isolate our main contributions.
>
> First, TOP-ERL is an episodic, essentially open-loop method: its policy takes only the initial state and outputs weights for movement primitives, which are combined with ProDMP to generate an entire trajectory that is then executed without replanning. By contrast, T-SAC is fully step-based and closed-loop, like standard SAC: at every time step the policy takes the current state as input and outputs the current action, allowing continuous reaction to intermediate states and disturbances.
>
> Second, the critics differ in both role and input. In TOP-ERL, the critic evaluates points along the planned MP trajectory. In T-SAC, the critic conditions directly on realized state–action sequences from replay (typically in joint space) generated by the closed-loop policy. Our sequence-conditioned Transformer critic thus estimates the value of actual state–action prefixes rather than abstract MP segments.
>
> We will add a concise comparison to the related-work section and soften the “bridging step-based and episodic regimes” phrasing. Our claim is that T-SAC keeps a standard step-based policy but augments it with a sequence-conditioned critic trained on long-horizon returns, which empirically works well on both long-horizon trajectory tasks (Box Pushing, Meta-World) and local-control tasks (MuJoCo).
>
> Regarding the reviewer’s concern about isolating the principal contribution: our ablations indicate that the Transformer critic trained with multi-step returns is the key driver of the gains. Comparing Fig. 5(b) (“min 1 max 1”) and Fig. 5(c) (“chunking length 1”) shows that even with step_length = 1 and soft target updates, T-SAC significantly outperforms SAC. Longer step_length and hard-copy target updates provide additional improvements only when combined with the Transformer critic: without the Transformer critic, the benefit of longer step_length is modest (Ablation (c)), and hard-copy updates are not applicable (otherwise one would need to resort to methods such as CrossQ). We will revise the ablation section to make this hierarchy explicit: the main novelty is the sequence-conditioned Transformer critic with $N$-step returns inside an off-policy actor–critic framework, while causal masking, parameter freezing, and $N$-step averaging are supporting design choices for stability and efficiency.
>
> >2. Comparisons with traditional off-policy N-step methods using IS are empirical only; formal connections to prior analyses are not developed.
> We appreciate the reviewer’s comment and agree that our treatment of off-policy N-step methods is primarily empirical in the current draft. Our focus in this work is on the architectural change (sequence-conditioned critic) and its practical behavior, but we will revise the paper to better connect our update to existing analyses of off-policy multi-step returns with IS.
>
> Concretely, our N-step target can be written as a standard N-step TD return with a single bootstrap at depth N, but without per-decision importance ratios. This places T-SAC in the same family as truncated/trace-cutting methods that deliberately trade strict unbiasedness for lower variance and better stability under function approximation. We will add a short subsection (and appendix note) that:
> (i) explicitly rewrites our critic update in the classical multi-step TD form;
> (ii) discusses its bias–variance trade-off relative to IS-corrected N-step returns under policy lag and UTD = 1; and
> (iii) links it to existing convergence and error-bound results for related off-policy multi-step schemes.
>
> We will also clarify in the text that our goal is not to introduce a new IS estimator, but to show that combining a sequence-conditioned critic with a simple, low-variance N-step target yields strong empirical performance. A full extension of existing theoretical frameworks to Transformer-based, sequence-conditioned critics is nontrivial and we view it as complementary future work.

---

> ### Author Response · Authors · 2025-12-02
>
> >3. Reported instabilities on Box-Pushing-Sparse lead to a fallback on soft targets, limiting generality. The action-mask requirement and performance variability on Ant/Hopper/Walker2d suggest fragility and sensitivity to dynamics/constraints.
>
> >4. T-SAC trails CrossQ on Hopper and Walker2d; the explanation “action mask needed may degrade performance” lacks systematic investigation. On Box-Pushing-Sparse, success is 58%, below TOP-ERL’s 70%; the claim of being “competitive under sparse feedback” does not reconcile this gap.
>
> We thank the reviewer for raising these points about stability and robustness. As detailed in our general response to all reviewers, we found that the reported instabilities on Box-Pushing-Sparse and the deficits on Hopper/Walker2d were caused not by the hard-copy mechanism or the action mask, but by our original high-variance target construction (randomly selecting one of two candidate targets). With the corrected conservative choice (taking the minimum of the two values), hard-copy T-SAC is stable on Box-Pushing-Sparse without falling back to soft targets and reaches about $60%$ success, and on Gymnasium MuJoCo it now performs on par with or better than CrossQ on Hopper and Walker2d. In the revised version, we will clarify the role of the action mask as a purely implementation-level safeguard for multi-step targets, correct our earlier attribution of instability to the mask, and update the results and discussion on robustness and performance variability across Hopper/Walker2d and Box-Pushing-Sparse.
>
> >5. No ablations on freeze length $K$ or reuse factor $N_c$ are provided to delineate stability regions.
>
> We agree that the current version does not explicitly characterize how freeze length and reuse factor affect stability. In the revision, we will add an ablation study that systematically varies (i) the freeze length of the hard-copy critic and (ii) the reuse factor (implemented as the step_length / number of gradient updates per environment step) on Walker2d (MuJoCo) locomotion tasks. This will let us delineate the region in which T-SAC remains stable and performance is robust.
>
> >6. Equation (1) defines $G^{(n)}$ with $V_\phi(s_{t+n})$ but does not clarify whether this is a soft or standard value; later (Sec. 4.3) it states the critic estimates the standard (non-soft) action-value. The notation should reflect this distinction.
>
> We thank the reviewer for pointing out this notational ambiguity. In our implementation, all action-values used in T-SAC are standard (non-soft) action-values: the critic is always trained to approximate the expected discounted return without an entropy term. Equation (1) is therefore intended to define the standard action-value, and Sec. 4.3 is consistent with this.
>
> To avoid confusion, we will revise the notation and accompanying text as follows:
>
> - In Eq. (1), we will explicitly write $Q^\pi(s_t, a_t)$ and clarify in the sentence below the equation that this is the standard (non-soft) action-value, i.e., it does not include the entropy bonus.
>
> - We will remove or rephrase any wording that could be interpreted as using a soft value function and ensure that all occurrences of $Q$ and $V$ are consistently described as non-soft throughout the paper.
>
> We believe these changes will resolve the ambiguity and make the distinction between standard and soft values clear in the notation.

---

> ### Author Response · Authors · 2025-12-02
>
> >7. All benchmarks are simulated continuous-control tasks; the paper does not evaluate discrete action spaces, stronger partial observability (beyond action history), or real-robot settings that motivate the approach.
>
> We agree that our current evaluation is limited to simulated continuous-control tasks and will clarify this scope in the paper. Our focus on continuous actions stems from the fact that T-SAC is built as a drop-in replacement for the standard continuous-control SAC critic, which is the dominant use case in long-horizon robotics benchmarks. Extending our sequence-conditioned critic and N-step scheme to discrete-action variants of SAC should be feasible in principle (the critic would still operate on state–action sequences), but would require additional design choices and baselines that we view as orthogonal to the main contribution; we will highlight this as a concrete direction for future work.
>
> For partial observability, we now include a Walker2d variant in which all velocity components are removed from the observation. In this setting, training for T-SAC becomes unstable, indicating that our current design does not yet robustly handle strong partial observability despite conditioning the critic on short histories. We will make this limitation explicit in the paper and discuss extensions with belief-state or recurrent encoders, and possibly sequence-conditioned actors, as an important avenue for future work.
>
> Finally, we do not yet include real-robot experiments because we lack a robust hardware and safety setup for large-scale evaluation. Instead, we follow common practice by using widely adopted simulated benchmarks that capture long-horizon, multi-phase, and contact-rich behaviors relevant to robotics. We see sim-to-real transfer and on-hardware validation as important extensions of this work and will state this more clearly in the discussion.

---

> ### Author Response · Authors · 2025-12-03
>
> **Questions:**
> >1. Can T-SAC be evaluated on discrete-action domains (e.g., Atari) or more strongly partially observable settings?
>
> We thank the reviewer for raising this point. Our current evaluation is indeed restricted to simulated continuous-control tasks, and we will make this scope explicit in the paper.
>
> T-SAC is implemented as a drop-in replacement for the standard continuous-control SAC critic, which is the primary setting for long-horizon robotics benchmarks. Extending our sequence-conditioned critic and N-step training scheme to discrete-action domains (e.g., Atari) should be conceptually straightforward: the critic would still operate on state–action sequences, but plugged into a discrete-action variant of SAC or a closely related off-policy algorithm. Doing so properly would, however, require additional design choices (e.g., handling large discrete action spaces, appropriate exploration mechanisms) and a new set of baselines that we see as orthogonal to the main contribution of this work. We will highlight this as a concrete direction for future work.
>
> Regarding partial observability, we now go beyond fully observed settings. In the revision, we add ablations on Walker2d that inject noise into actions and states, introduce random terminations, and consider a partially observed variant where all velocity components are removed from the observation. In this partially observed Walker2d, T-SAC training becomes unstable, indicating that our current design does not yet robustly handle strong partial observability despite conditioning the critic on short histories. We will make this limitation explicit and discuss extensions with belief-state or recurrent encoders, and possibly sequence-conditioned actors, as an important direction for future work.
>
> >2. Do the benefits of sequence-conditioned critics carry over to image-based ML1 or partially observed Box-Pushing variants?
>
> We appreciate the reviewer’s question about image-based ML1 and partially observed Box-Pushing variants. In this work we intentionally focused on the standard, low-dimensional state settings used by closely related methods such as TOP-ERL, Simba-v2, CrossQ, and BroNet, which also do not report results on image-based ML1. Extending to pixel observations would require a substantial additional effort (choice and training of a visual encoder, stability tuning, and significantly higher compute), which is orthogonal to the question we target here: whether sequence-conditioned critics with multi-step returns improve long-horizon credit assignment in continuous-control benchmarks.
>
> Architecturally, our approach is compatible with image-based settings. The Transformer critic operates on sequences of feature vectors and is agnostic to the input modality; in an image-based ML1 configuration, one could feed it embeddings from a convolutional or ViT-style encoder (either frozen or jointly trained) together with actions and apply the same N-step training scheme. We will clarify this in the paper and add a short discussion outlining how T-SAC can be combined with standard visual encoders.
>
> For partially observed Box-Pushing variants, sequence conditioning is arguably even more natural: the critic can process histories of observations and actions and thereby approximate a belief-dependent value function. To fully exploit partial observability, however, one would ideally also equip the policy with memory (e.g., a recurrent or history-augmented actor) rather than keeping it purely Markovian. In the revision, we will clarify that our current experiments use fully observed Box-Pushing, and discuss how the same sequence-conditioned critic architecture can be directly applied to partially observed variants by operating on observation–action histories, which we view as a natural direction for future work.

---

> ### Author Response · Authors · 2025-12-03
>
> >3. How does T-SAC compare with Transformer-based policies and offline RL approaches (e.g., Decision Transformer)?
>
> We agree that it is important to position T-SAC relative to Transformer-based policies and offline RL methods such as Decision Transformer, and we will clarify this more explicitly in the revised manuscript.
>
> **Transformer-based policies**. Our experiments already include a Transformer-based policy baseline in the form of a GTrXL-based recurrent policy, which places the Transformer on the policy side. By contrast, T-SAC keeps a standard SAC policy and only replaces the critic with a sequence-conditioned Transformer trained with N-step returns. We also compare against TOP-ERL, which uses a Transformer on the critic side in an episodic, open-loop control setting. Empirically, T-SAC consistently outperforms both the GTrXL policy baseline and TOP-ERL on our long-horizon benchmarks, including Meta-World ML1 and Box Pushing, as well as on standard MuJoCo control. We will highlight these comparisons more clearly in the experimental section and add a short table in the related-work section emphasizing the distinction between “Transformer-in-the-critic” (T-SAC and TOP-ERL, but with different control regimes) and “Transformer-in-the-policy” (GTrXL baseline).
>
> **Offline RL / Decision Transformer**. Decision Transformer and related methods operate in the offline RL setting, where a fixed dataset of trajectories is given and no further environment interaction is allowed, while T-SAC is designed for online off-policy RL. A direct empirical comparison would therefore require switching to an offline evaluation protocol (e.g., D4RL-style datasets or offline variants of Meta-World), which is orthogonal to our current focus on online learning. For the benchmarks we consider (MuJoCo, Box Pushing, and Meta-World ML1 in the standard online setting), there is no commonly adopted offline dataset and evaluation protocol matching our setup. Rather than mixing settings, we chose to compare against strong online off-policy baselines (SAC, CrossQ, GTrXL, TOP-ERL). We will add a paragraph to the related-work section explicitly discussing Decision Transformer and offline RL as complementary approaches—using sequence modeling at the policy level on fixed datasets—whereas T-SAC uses sequence modeling in the critic to improve temporal credit assignment in an online, off-policy actor–critic framework.
>
> >4. For GRU/LSTM critics, how extensive was the hyperparameter and architecture search (depth, hidden size, normalization, teacher forcing, sequence length)?
>
> We thank the reviewer for raising this question. For the GRU/LSTM critics we performed a targeted hyperparameter search over network depth and hidden size. In particular, we varied the number of recurrent layers and the width of the hidden state, and selected the best configuration on the validation performance.
>
> We did not perform an additional sweep over normalization or teacher forcing; these settings were kept fixed across all GRU/LSTM critics to keep the comparison computationally feasible and to focus the comparison on architectural differences rather than extensive per-model tuning.
>
> The sequence length for GRU/LSTM critics was set to be the same as the default sequence length used by T-SAC, so that all sequence-based critics operate over comparable horizons. We will clarify these design choices and the scope of the hyperparameter search in the revised manuscript, and we acknowledge that a broader search over recurrent-model-specific tricks (e.g., normalization, teacher forcing) may further strengthen these baselines.

---

> ### Author Response · Authors · 2025-12-04
>
> > 5. Have you tested robustness under observation/action noise, stochastic terminations, or mixed-policy replay buffers? This seems particularly relevant given the “off-policy without IS” choice.
>
> We agree that robustness under additional sources of stochasticity is an important question, especially given our choice to omit importance sampling (IS), and we appreciate the reviewer raising it. In the revision, we will add dedicated robustness ablations on Walker2d that explicitly inject (i) observation noise, (ii) action noise, and (iii) stochastic early terminations. We will also treat a partially observed Walker2d variant—where all velocity components are removed from the observation—as an extreme robustness test; in this setting, T-SAC training becomes unstable, and we will explicitly document this as a limitation of the current design.
>
> Our existing benchmarks (Box Pushing, Meta-World, MuJoCo) already involve (i) stochastic initial states, (ii) exploration noise in the actor, and (iii) replay buffers populated by a continually evolving policy, so T-SAC is already trained on a non-stationary, mixed-policy buffer. We will clarify this in the text and leave more aggressive buffer-mixing schemes (e.g., replaying data from much older policies) to future work.
>
> Regarding the “off-policy without IS” choice: importance sampling is introduced when one wants to estimate expectations under the current policy using data from a different behavior policy. In our implementation, the critic update is defined on specific state–action pairs sampled from replay: given a state s and action a, the reward and next state are determined by the environment dynamics and do not depend on how likely a is under the current policy. As in SAC/TD3, we therefore omit IS corrections, trading strict unbiasedness with respect to the current policy for lower-variance updates, which is particularly important when combining multi-step returns with a large function approximator such as a Transformer critic.
>
> >6. What aspects of the design (e.g., window sampling, bootstrap lag, prefix length) most limit performance in sparse-reward settings, and could targeted modifications (e.g., auxiliary returns, adaptive horizons) mitigate the observed instability?
>
> We thank the reviewer for this question. As discussed in our first comment: the observed instability in sparse-reward tasks was due to an implementation issue rather than limits of window sampling, bootstrap lag, or prefix length. With the corrected implementation, T-SAC is stable and competitive on Box-Pushing Sparse and other Mujoco benchmarks, and we will clarify this in the revision.
>
> [1]Li, Qiyang, Zhiyuan Zhou, and Sergey Levine. "Reinforcement learning with action chunking." arXiv preprint arXiv:2507.07969 (2025).
>
> [2]Li, Ge, et al. "Top-erl: Transformer-based off-policy episodic reinforcement learning." arXiv preprint arXiv:2410.09536 (2024).
>
> [3]Parisotto, Emilio, et al. "Stabilizing transformers for reinforcement learning." International conference on machine learning. PMLR, 2020.

---

### Author Response · Authors · 2025-11-27
**Official comment to all reviewers: stability, target construction, and action masks.**

We thank all reviewers for their thoughtful, professional, and precise feedback, which we found invaluable for improving both the clarity and rigor of our work. Several reviewers also raised two closely related concerns: (i) the reported instabilities on Box-Pushing-Sparse, which led us to fall back to soft targets and seemed to limit the generality of hard-copy, and (ii) the apparent dependence on action masks and resulting sensitivity to the dynamics and constraints in Hopper and Walker2d.

After further investigation, we found that these issues were not caused by the hard-copy mechanism or the Transformer critic, but by our particular choice of target value in the original submission. Specifically, we defined the target by randomly selecting one of two candidate target values. Our motivation was that this gives an unbiased estimator of the target, whereas using the minimum of the two values is pessimistic and systematically underestimates it. Preliminary experiments suggested that this unbiased estimator could accelerate convergence, which is why we adopted it in the submitted version.

However, this design is problematic for our optimization problem. In standard trust-region optimization for maximization [1], a common safeguard is to require that the surrogate objective underestimate or correctly estimate the true objective, so that updates are conservative (safe) rather than overly aggressive. Even though we do not use a trust-region update in our setting, the requirement on the surrogate objective remains the same because the critic network is another kind of surrogate function. By randomly choosing between two targets, our estimator sometimes overestimates and sometimes underestimates the true value; the overestimation events can lead to unsafe updates and thus instability. This is consistent with the argument in TOP-ERL [2] that using the minimum of two target values is typically preferable to using their mean; our random-choice scheme is even less conservative than the mean and has higher variance. The same reasoning about higher variance also underlies the argument in App. D.

This mechanism also explains the occasional failures on Hopper/Walker2d. Random target overestimation can trigger unsafe policy updates which, in these environments, often manifest as early episode termination, making performance appear highly sensitive to the underlying dynamics and constraints.

Our original explanation ("the action mask may degrade performance") was therefore misleading. Subsequent experiments show that the main culprit is not the action mask itself, but this high-variance target-selection scheme. In environments where an action mask is used (Hopper, Walker2d), unsafe updates caused by target overestimation more easily lead to catastrophic failures, so the problem appeared to correlate with the presence of action masks. Because we initially observed instability only in masked environments, we incorrectly attributed the performance drop to the mask rather than to the target construction.

With the corrected choice of target—using the minimum of the two candidate target values, as motivated by trust-region arguments [1], TOP-ERL [2], and App. D—the instability disappears while keeping the action mask fixed. Hard-copy T-SAC now performs on par with or better than CrossQ on the MuJoCo locomotion tasks (including Hopper and Walker2d), indicating that the action mask does not inherently degrade performance; the earlier deficits were a side effect of our high-variance, occasionally overestimating target.

On Box-Pushing-Sparse, the original 58% success rate was indeed below TOP-ERL’s reported 70%, so our claim of being “competitive under sparse feedback” was conservative at best. Under the corrected minimum-based target scheme, hard-copy T-SAC reaches about 60% success on Box-Pushing-Sparse, which is more competitive with TOP-ERL in the same sparse-reward setting.

We will update the manuscript to: (i) replace the random target-selection rule with the conservative minimum-based target; (ii) report the revised results on Hopper/Walker2d and Box-Pushing-Sparse.

[1] Nocedal, Jorge, and Stephen J. Wright. Numerical Optimization. New York, NY: Springer New York, 2006.

[2] Li, Ge, et al. "TOP-ERL: Transformer-based off-policy episodic reinforcement learning." arXiv preprint arXiv:2410.09536 (2024).

---

### Meta-Review · Area_Chair_jfLs · 2026-01-09

**Summary:**

The primary concerns informing the decision centered on the method's perceived instability and lack of robustness across different domains, specifically the reported performance degradation on standard MuJoCo tasks like Hopper and Walker2d compared to baselines, and the inability to use the proposed "hard-copy" update scheme in sparse-reward settings without falling back to soft targets. Reviewers questioned the generality of the approach, suspecting that the reliance on action masks masked underlying fragility, and sought stronger theoretical justifications for the Transformer-based critic design and the omission of importance sampling in the N-step updates, alongside requests for clearer differentiation from prior work like TOP-ERL and broader evaluations in discrete or partially observable settings.

**Reviewer Concerns:**

The rebuttal successfully addressed the most critical concerns regarding stability and performance deficits by identifying that the root cause was not the method itself but a specific, flawed implementation choice of randomly selecting target values; the authors demonstrated that switching to a standard conservative minimum-target update resolved the instability on Box-Pushing-Sparse and recovered performance on Hopper and Walker2d, thereby validating the "hard-copy" mechanism and clarifying that the action mask is merely a routine boundary safeguard. The authors also effectively distinguished their closed-loop approach from the open-loop TOP-ERL and provided intuitive explanations for the benefits of gradient averaging over target averaging, although concerns regarding the lack of extensive evaluation on high-dimensional visual inputs or discrete action spaces remain largely outstanding as the authors positioned these as future work, providing only limited ablations for partial observability.

**Reviewer Scores:**

It is highly probable that Reviewers qLQQ and 9zeX would raise their scores from just below the threshold to a clear acceptance, as their primary grounds for rejection—specifically the method's fragility, the underperformance on standard locomotion tasks, and the apparent need for "hacks" like soft targets in sparse settings—were directly negated by the correction of the target-selection bug and the subsequent improved empirical results. Reviewers m1ZH and 2mBX, who were already marginally positive, would likely increase their confidence and scores further, seeing that the proposed target-free freezing schedule is now shown to be robust across both dense and sparse reward settings, effectively removing the main caveat they had placed on the paper's contribution.

---

### Decision · Program_Chairs · 2026-01-26

Accept (Poster)